# Cerebrovascular reactivity assessment with O$_2$-CO$_2$ exchange ratio under brief breath hold challenge

Suk-tak Chan[1]*, Karleyton C. Evans[2,¤a], Tian-yue Song[1,2], Juliette Selb[1,¤b], Andre van der Kouwe[1], Bruce R. Rosen[1], Yong-ping Zheng[3], Andrew Ahn[1], Kenneth K. Kwong[1]

**1** Athinoula A. Martinos Center for Biomedical Imaging, Department of Radiology, Massachusetts General Hospital, Charlestown, Massachusetts, United States of America, **2** Department of Psychiatry, Massachusetts General Hospital, Charlestown, Massachusetts, United States of America, **3** Department of Biomedical Engineering, The Hong Kong Polytechnic University, Hong Kong Special Administrative Region, China

¤a Current address: Biogen Inc., Cambridge, Massachusetts, United States of America
¤b Current address: ODS Medical, Inc., Montreal, Quebec, Canada
* phoebe@nmr.mgh.harvard.edu

**Data Availability Statement:** All relevant data without subject identifiers are within the manuscript and its Supporting Information files. Individual imaging data with subject identifiers

## Abstract

### Background

Hypercapnia during breath holding is believed to be the dominant driver behind the modulation of cerebral blood flow (CBF). However, increasing evidence show that mild hypoxia and mild hypercapnia in breath hold (BH) could work synergistically to enhance CBF response. We hypothesized that breath-by-breath O$_2$-CO$_2$ exchange ratio (bER), defined as the ratio of the change in partial pressure of oxygen ($\Delta PO_2$) to that of carbon dioxide ($\Delta PCO_2$) between end inspiration and end expiration, would be able to better correlate with the global and regional cerebral hemodynamic responses (CHR) to BH challenge. We aimed to investigate whether bER is a more useful index than end-tidal PCO$_2$ to characterize cerebrovascular reactivity (CVR) under BH challenge.

### Methods

We used transcranial Doppler ultrasound (TCD) to evaluate CHR under BH challenge by measuring cerebral blood flow velocity (CBFv) in the middle cerebral arteries. Regional changes in CHR to BH and exogenous CO$_2$ challenges were mapped with blood oxygenation level dependent (BOLD) signal changes using functional magnetic resonance imaging (fMRI). We correlated respiratory gas exchange (RGE) metrics (bER, $\Delta PO_2$, $\Delta PCO_2$, end-tidal PCO$_2$ and PO$_2$, and time of breaths) with CHR (CBFv and BOLD) to BH challenge. Temporal features and frequency characteristics of RGE metrics and their coherence with CHR were examined.

### Results

CHR to brief BH epochs and free breathing were coupled with both $\Delta PO_2$ and $\Delta PCO_2$. We found that bER was superior to either $\Delta PO_2$ or $\Delta PCO_2$ alone in coupling with the changes of

cannot be shared publicly because of institutional policies regarding to data sharing and protection of research subject privacy. The IRB protocols and the consent under which the subjects received imaging scans did not include language which permits the inclusion of their images in public data repositories. Data are available for researchers who meet the criteria from Massachusetts General Hospital IRB of Partners HealthCare. Researchers seeking to utilize the de-identified imaging data from this manuscript should contact the corresponding author (phoebe@nmr.mgh.harvard.edu) and Partners Research Management (mghsubs@partners.org) for a Data Use Agreement.

**Funding:** This research was carried out in whole at the Athinoula A. Martinos Center for Biomedical Imaging at the Massachusetts General Hospital, using resources provided by the Center for Functional Neuroimaging Technologies, P41EB015896, a P41 Biotechnology Resource Grant supported by the National Institute of Biomedical Imaging and Bioengineering (NIBIB), National Institutes of Health, as well as the Shared Instrumentation Grant S10RR023043. This work was also supported, in part, by NIH-K23MH086619.

**Competing interests:** The authors have declared that no competing interests exist.

CBFv and BOLD signals under breath hold challenge. The regional CVR results derived by regressing BOLD signal changes on bER under BH challenge resembled those derived by regressing BOLD signal changes on end-tidal $PCO_2$ under exogenous $CO_2$ challenge.

## Conclusion

Our findings provide a novel insight on the potential of using bER to better quantify CVR changes under BH challenge.

## Introduction

Breath hold challenge has been used in the clinical setting as a simple vasoactive stimulus for the assessment of cerebrovascular reactivity (CVR) [1, 2] in patients with carotid artery diseases [3–5] as well as brain tumors [6]. Its use for CVR assessment with transcranial Doppler sonography (TCD) measurement was first demonstrated by Ratnatunga and Adiseshiah [3]. The cerebral blood flow velocities (CBFv) were often measured in the middle cerebral arteries which supply most parts of the brain. The time of breaths (ToB) had often been taken as an indicator of the strength of the vasoactive stimulus to induce changes of cerebral blood flow (CBF). The ratio of CBF to ToB was known as a breath hold index to evaluate CVR [1, 2, 4, 5, 7]. Although TCD offers high temporal resolution to evaluate cerebrovascular responses without the concern of aliasing high frequency hemodynamic signal into the low frequency range, it does not provide regional information. Therefore regional CVR mapping with blood oxygen level-dependent (BOLD) signal changes measured by functional magnetic resonance imaging (fMRI) was used. BOLD-fMRI was used instead of arterial spin labeling (ASL) in MRI perfusion studies due to the low contrast to noise ratio and the low temporal resolution of the ASL technique [8]. ASL image acquisition at a temporal resolution of 4 seconds will under-sample the brain responses within respiratory cycle of 4–6 seconds.

Hypercapnia or increased end-tidal partial pressure of $CO_2$ ($P_{ET}CO_2$) was commonly measured as a surrogate for the increased arterial $CO_2$ level to evaluate CVR [9–11]. Hypoxia or changes of end-tidal partial pressure of $O_2$ ($P_{ET}O_2$) were seldom used to account for CBF changes during breath holding partly due to the common belief that the vasodilatory effect of increased arterial partial pressure of $CO_2$ ($PCO_2$) dominates that of decreased arterial partial pressure of $O_2$ ($PO_2$). Such a belief stemmed from high altitude studies and lab-controlled low oxygen environment reporting that significant CBF changes only happened in hypoxia with arterial $PO_2$ ($PaO_2$) going down to approximately 50 mmHg [12–14]. It should be noted that reports of large CBF increase in response to strong hypoxia were often accompanied by hypocapnia due to hyperventilation in the studies of either high altitudes or lab-controlled low oxygen environment (~8–13% $O_2$) [12–14]. The case is different with breath hold because hypoxia is accompanied by hypercapnia and not by hypocapnia during breath holding.

The role of both hypoxia and hypercapnia in breath hold deserves to be examined because arterial $PO_2$ and $PCO_2$ had been reported to work synergistically at *normoxia* to interact with peripheral chemoreceptors [15–19]. In spontaneous breathing, blood gas levels of $O_2$ and $CO_2$ are optimized by the feedback control of ventilation via chemoreflexes [20] to regulate blood flow and oxygen delivery to the brain as part of a vital homeostatic process [17]. The feedback loops include interaction between central chemoreceptors at the brainstem [21, 22] and peripheral chemoreceptors at the carotid body [23, 24]. While most of the studies on the role of $PO_2$ to stimulate peripheral chemoreceptors at the carotid body had been focused on strong hypoxia [17] where chemoreceptor activities rose quickly in a hyperbolic fashion, Biscoe et al.

[15] showed that peripheral chemoreceptor activities could be observed from normoxia to hyperoxia up to arterial $PO_2$ level of 190mmHg and beyond. Lahiri et al. [18] reported that the stimulus thresholds of arterial $PO_2$ and $PCO_2$ for peripheral chemoreceptors were largely interdependent under the normoxic condition where a drop in arterial $PO_2$ was routinely accompanied by increased chemoreceptor activities as well as an enhanced sensitivity of carotid chemoreceptors to arterial $PCO_2$. Several authors [15–19] reported similar findings for normoxic as well as hypoxic conditions. Ventilation, like chemoreceptors, also becomes more sensitive to $PCO_2$ with a slight decrease in $PO_2$ within the range of normoxia (90–110 mmHg). The same interaction between change in $PO_2$ and change in peripheral chemoreceptor activities during spontaneous breathing is also assumed to take place during breath hold.

In terms of the relationship between modulated chemoreceptor activities and cerebral hemodynamic responses, previous studies reported that apnea-induced hypoxia and hypercapnia caused chemoreceptor-mediated central vasodilation and concurrent peripheral vasoconstriction to conserve oxygen delivery to the brain [25], leading to an increase in cerebral blood flow (CBF) and a decrease in peripheral oxygen saturation [26]. Enhanced hemodynamic responses to mild hypoxia with maintained eucapnia or in the presence of hypercapnia in humans had also been demonstrated in the work by Shapiro et al. [27] and Mardimae et al. [28]. Those authors reported progressive CBF increase in response to small steps of serial reductions of $PO_2$ starting from normoxia with maintained eucapnia [27] as well as with slight hypercapnia of around 45 mmHg [28]. The findings in these studies suggested that mild hypercapnia could increase the sensitivity of the CBF response to a very mild level of hypoxia and the ranges of mild $PO_2$ and $PCO_2$ changes reported are achievable by breath hold.

Beyond the belief of $CO_2$ being the dominant effect to account for CBF changes, the perception that effect of $O_2$ on CBF simply reflects the effect of $CO_2$ is often associated with another belief that changes of $PO_2$ and $PCO_2$ in respiration are mirror image of each other [29]. However, respiratory data acquired on 12 subjects in the study by Lenfant [30] showed that the time courses of $O_2$ and $CO_2$ were interdependent but not redundant in terms of their temporal and frequency characteristics. Even without being redundant, such interdependence can still pose some problems for CVR analysis. There is a question of collinearity when interdependent changes of $PCO_2$ and $PO_2$ are included as predictor variables in the same regression model of CVR quantification. We are seeking a combination of $PO_2$ and $PCO_2$ that can properly characterize the interdependence between $PO_2$ and $PCO_2$ without creating a problem of collinearity in CVR analysis. The use of the breath-by breath $O_2$-$CO_2$ exchange ratio described in the alveolar air equation [31, 32] is a natural target to be investigated for breath hold challenge in the regression model.

We preferred the terms $\Delta PO_2$ (inspired $PO_2$ –expired $PO_2$) and $\Delta PCO_2$ (expired $PCO_2$ – inspired $PCO_2$) over the more commonly used $P_{ET}O_2$ and $P_{ET}CO_2$ because the breath-by-breath $\Delta PO_2$ and $\Delta PCO_2$ are the terms used in the alveolar air equation [31, 32] to describe the change of gas partial pressure in systemic $O_2$ uptake and that in $CO_2$ release respectively. We also prefer to use $\Delta PO_2/\Delta PCO_2$ as our breath-by-breath $O_2$-$CO_2$ exchange ratio (bER). bER takes advantage of its ratio format to reduce the unwanted effects of ventilatory volume fluctuations due to isolated deep breaths common to $\Delta PO_2$ and $\Delta PCO_2$ measured. bER is mathematically equivalent to the *reciprocal* of the respiratory exchange ratio (RER) from alveolar air equation [31, 32], except that bER is a dynamic breath-by-breath measurement and RER is a steady-state time-averaged measurement over a period of time. RER has been used to evaluate resting systemic metabolic rate [33–35]. Some technical differences do need to be mentioned between RER used in the literature and bER we used in this study. Traditionally, RER is derived by measuring the respiratory flow and the expired gases collected in Douglas bag

connected to a closed circuit over several minutes. bER is derived here by measuring the inspired and expired gases with a nasal tubing at each breath.

In this study, we hypothesized that mild hypoxia and hypercapnia work synergistically to increase CBF under breath hold challenge. The primary objective was to evaluate our hypothesis that bER would be a more useful index than $P_{ET}CO_2$ to correlate with the changes of CBFv and BOLD signals in the evaluation of cerebral hemodynamic responses to the breath hold challenge. To address the question of redundancy among respiratory gas exchange (RGE) metrics, we examined the correlations among bER, $\Delta PCO_2$, $\Delta PO_2$, $P_{ET}CO_2$, $P_{ET}O_2$ and ToB in both TCD and MRI sessions. In the first part of the study, we correlated the time series of the RGE metrics (bER, $\Delta PCO_2$, $\Delta PO_2$ and ToB) with those of CBFv and BOLD signal changes under the protocol of breath hold challenge. We then examined the temporal features and frequency characteristics of these RGE metrics as well as their coherence with CBFv and BOLD signal changes. In the second part of the study, we compared the regional CVR maps obtained by regressing BOLD signal changes on selected RGE metrics including bER, $P_{ET}CO_2$ and ToB under breath hold challenge with regional CVR maps obtained by regressing BOLD signals changes on $P_{ET}CO_2$ obtained under exogenous $CO_2$ hypercapnic challenge. The success of such association between bER and cerebral hemodynamic responses, in addition to offering a better physiological model to characterize cerebral hemodynamic responses under breath hold challenge, would also provide a novel insight in the study of brain-body interaction.

## Materials and methods

### Participants

Seventeen volunteers aged from 22 to 48 years (mean = 31.18 years; SD = 8.78 years; 11M and 6F) were included. All of them were recruited by e-mail and poster placement within the Partners hospital network. They were screened to exclude neurological, mental and medical disorders and drug abuse. TCD and MRI scanning were performed in the Athinoula A. Martinos Center for Biomedical Imaging at the Massachusetts General Hospital of Partners HealthCare. All the experimental procedures were explained to the subjects, and signed informed consent was obtained prior to participation in the study. All components of this study were performed in compliance with the Declaration of Helsinki and all procedures were approved by Partners Human Research Committee.

Our study was divided into Part 1 and Part 2. In Part 1, we aimed to correlate the RGE metrics including bER, $\Delta PO_2$, $\Delta PCO_2$ and ToB with cerebral hemodynamic responses including CBFv and BOLD signal changes to breath hold challenge. We also examined the temporal features and frequency characteristics of RGE metrics and their coherence with cerebral hemodynamic responses. Sixteen out of 17 subjects performed breath hold tasks in the MRI sessions, while 12 of them participated in TCD sessions. In Part 2, we aimed to assess the usefulness of bER in the regional breath hold CVR quantification by comparing the regional CVR maps obtained by regressing BOLD signal changes on bER, $P_{ET}CO_2$ and ToB under breath hold challenge with regional CVR maps obtained by regressing BOLD signals changes on $P_{ET}CO_2$ under exogenous $CO_2$ challenge. Ten out of 17 subjects had additional exogenous $CO_2$ challenge for comparison. Before we correlated the changes of RGE metrics with CBFv and BOLD signal changes under breath hold challenge, we examined the correlations among the RGE metrics (bER, $\Delta PCO_2$, $\Delta PO_2$, $P_{ET}CO_2$, $P_{ET}O_2$ and ToB) acquired in both TCD and MRI sessions.

## Part 1: Breath hold challenge

**Transcranial Doppler scanning.** Before the study of blood flow velocity in intracranial arteries, subject was allowed to rest at least 20–30 minutes for hemodynamic stabilization. The blood pressure measured in the subject was within the normal range [36]. With the subject in a sitting position, a dual probe setting with 2MHz transducers in conjunction with TCD system (Delicate EMS-9U, Shenzhen, China) was used for simultaneous recording of CBFv in the middle cerebral arteries (MCA) on both left and right sides while the subject was performing the breath hold task. Two transducers were attached onto the left and right temporal bone window by velcro. The depth of the Doppler samples was confined to the M1 segment, which is at the main stem of the MCA, for all the subjects.

The timing of the breath hold task was presented visually to the subject by a computer using the software Eprime Professional 2.0 (Psychology Software Tools, Inc., Pittsburgh, USA). A rehearsal session was given to each subject to practice breath hold task. Each subject was instructed via visual cues to perform 6 epochs of 30-second breath hold interleaved with 60–90 seconds of free breathing (Fig 1). They were instructed by visual cues to only hold their breath for as long as they could during the 30-second period. Multiple epochs of breath holding followed by free breathing increased the samples for quantitative analysis. The total duration of the breath hold protocol lasted 10 minutes.

Physiological changes including $PCO_2$, $PO_2$, electrocardiogram (ECG) and peripheral blood pressure were measured simultaneously with TCD acquisition. A small nasal tubing was placed at the subject's nostril to sample $PCO_2$ and $PO_2$ via gas analyzers (Capstar-100, Oxystar-100, CWE, Inc., PA, USA) after calibrating to the barometric pressure of the day of TCD scanning and correcting for vapor pressure. Peripheral blood pressure was continuously measured with Finometer MIDI (Finapres Medical Systems B.V., Netherlands). All the TCD and physiological measurements were synchronized using trigger signals from E-prime. CBFv time series and physiological recordings were stored for offline data analysis.

**MRI acquisition.** MRI brain scanning was performed on a 3-Tesla scanner (Siemens Medical, Erlangen, Germany). The head was immobilized in a standard head coil with foam pads. The following whole brain MRI datasets were acquired on each subject: 1) standard high-resolution sagittal images acquired with volumetric T1-weighted 3D-MEMPRAGE sequence (TR = 2530ms, TE = 1.74ms/3.6ms/5.46ms/7.32ms, flip angle = 7˚, FOV = 256×256mm, matrix = 256×256, slice thickness = 1mm); 2) BOLD-fMRI images acquired with gradient-echo echo planar imaging (EPI) sequence (TR = 1450ms, TE = 30ms, flip angle = 90˚, FOV = 220×220mm, matrix = 64×64, slice thickness = 5mm, slice gap = 1mm) while the subject was performing the breath hold task. The breath hold task and the physiological set-up used for gas sampling in MRI session were the same as those used in TCD sessions. The gas analyzers were again calibrated to the barometric pressure of the day of MRI scanning and corrected for vapor pressure. ECG was measured using Siemens physiological monitoring unit. Physiological changes including $PCO_2$, $PO_2$ and ECG were measured simultaneously with MRI acquisition. All the physiological measurements were synchronized using trigger signals from the MRI scanner. BOLD-fMRI images and physiological recordings were stored for offline data analysis.

## Part 2: Exogenous $CO_2$ challenge

Ten out of 16 subjects had additional exogenous $CO_2$ challenge in the MRI session. Given that there is significant inter-individual variance in resting $P_{ET}CO_2$ [37], resting $P_{ET}CO_2$ was assessed in those subject via calibrated capnograph before the exogenous $CO_2$ challenge. Subject wore nose-clip and breathed through a mouth-piece on an MRI-compatible circuit

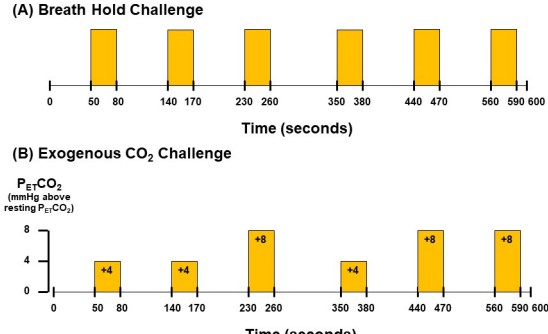

**Fig 1.** Paradigms of (A) breath hold and (B) exogenous $CO_2$ challenges. The timing for the onset and the end of breath hold epochs is the same as that for exogenous $CO_2$ hypercapnic epochs. The duration of normal breathing phases in breath hold challenge and normocapnic phases in exogenous $CO_2$ challenge varies from 60 seconds to 90 seconds.

designed to maintain the $P_{ET}CO_2$ within ± 1–2 mmHg of target $P_{ET}CO_2$ [38, 39]. The fraction of inspired carbon dioxide was adjusted to produce steady-state conditions of normocapnia and mild hypercapnia (4–8 mmHg above the subject's resting $P_{ET}CO_2$). The $CO_2$ challenge paradigm consisted of 2 consecutive phases (normocapnia and mild hypercapnia) repeating 6 times with 3 epochs of 4 mmHg increase and 3 epochs of 8 mmHg increase of $P_{ET}CO_2$ (Fig 1). The normocapnia phase lasted 60–90 seconds, while the mild hypercapnia phase lasted 30 seconds. The total duration of the exogenous $CO_2$ hypercapnic challenge lasted 10 minutes.

When the subject had exogenous $CO_2$ challenge in MRI session, BOLD-fMRI images were acquired using the same EPI sequence for breath hold challenge. The $PCO_2$ and $PO_2$ were sampled through the air filter connected with the mouthpiece and the sampled gases were measured by calibrated gas analyzers. The respiratory flow was measured with respiratory flow head (MTL300L, AdInstruments, Inc., CO, USA) on the breathing circuit via calibrated spirometer (FE141, AdInstruments, Inc., CO, USA). The physiological measurements were synchronized with MRI images using trigger signals from MRI scanner. All the BOLD-fMRI images and physiological recordings were stored for offline data analysis.

## Data analysis

**Processing of physiological data.** The physiological data from both TCD and MRI sessions were analyzed using Matlab R2014a (Mathworks, Inc., Natick, MA, USA). Technical delay of $PCO_2$ and $PO_2$ was corrected by cross-correlating the time series of $PCO_2$ and $PO_2$ with respiratory phases determined from the artifactual displacement due to chest excursion on ECG time series for the breath hold runs in the TCD sessions, with the respiratory phases from the respiratory bellow for the breath hold runs in the MRI sessions, and with the respiratory flow for the exogenous $CO_2$ runs.

End inspiration (I) and end expiration (E) were defined on the time series of $PO_2$ and $PCO_2$ (S1 Fig). They were verified by the inspiratory and expiratory phases on the respiration time series. The breath-by-breath end-tidal $CO_2$ ($P_{ET}CO_2$) and end-tidal $O_2$ ($P_{ET}O_2$) were extracted at the end expiration of $PCO_2$ and $PO_2$ time series respectively. Changes of the gas parameters during breath hold periods were interpolated by the values measured immediately before and after the breath hold periods. The duration of breathing cycle, which is known as time of breath (ToB), was derived by subtracting the timing in seconds of the 2 consecutive end expiration markers. Breath-by-breath $O_2$-$CO_2$ exchange ratio (bER) is defined as the ratio of the change in $PO_2$ ($\Delta PO_2$ = inspired $PO_2$ –expired $PO_2$) to the change in $PCO_2$ ($\Delta PCO_2$ =

expired $PCO_2$ –inspired $PCO_2$) measured between end inspiration and end expiration in each respiratory cycle.

Simple correlation analyses were applied on the time series of RGE metrics (bER, $\Delta PCO_2$, $\Delta PO_2$, $P_{ET}CO_2$, $P_{ET}O_2$ and ToB) in pairs, which were acquired in both TCD and MRI sessions. Time series of the RGE metrics including bER, $\Delta PO_2$, $\Delta PCO_2$ and ToB were also used in the CBFv and BOLD data analyses.

**Preprocessing of CBFv data.** The CBFv data were analyzed using Matlab R2014a (Mathworks, Inc., Natick, MA, USA). A median filter was applied to the data to reduce artifactual spikes. Beat-by-beat systolic peaks and end-diastolic troughs were determined using custom Matlab function and corrected on the graphical user interface incorporated in the function. Systolic peaks and end-diastolic troughs of cardiac cycles on the CBFv time series showing persistent artifacts were excluded in the following analysis. TCD data in both left and right MCAs were acquired on 12 subjects. One of the 12 TCD datasets had persistent artifacts in over one-third of the time series acquired in the LMCA. The CBFv time series in the LMCA of that particular TCD dataset was excluded in further analysis. Time series of mean CBFv was derived by averaging the CBFv over each cardiac cycle. In order to reduce the large inter-individual variations of absolute blood flow velocities [40, 41] and to remove the dependence of insonation angle [42], the percent change of CBFv ($\Delta$CBFv) of the left and right MCAs relative to baseline value was derived. The mean CBFv for a period of 30 seconds at the beginning of the time series was chosen as the baseline because the subject was in resting state and the CBFv acquired within this period had the least effect from the respiratory challenges.

**Preprocessing of BOLD-fMRI data.** All the BOLD-fMRI data were imported into the software Analysis of Functional NeuroImage (AFNI) [43] (National Institute of Mental Health, http://afni.nimh.nih.gov) for time-shift correction, motion correction, normalization and detrending. The first 12 volumes in the first 12 time points of each functional dataset, collected before equilibrium magnetization was reached, were discarded. Each functional dataset was corrected for slice timing, motion-corrected and co-registered to the first image of the first functional dataset using three-dimensional volume registration. It was then normalized to its mean intensity value across the time-series. Voxels located within the ventricles and outside the brain defined in the parcellated brain volume using FreeSurfer [44] (MGH/MIT/HMS Athinoula A. Martinos Center for Biomedial Imaging, Boston, http://surfer.nmr.mgh.harvard. edu) were excluded from the following analyses of functional images. The time-series of each voxel in the normalized functional dataset was detrended with the $5^{th}$ order of polynomials to remove the low drift frequency. Individual brain volumes with time series of percent BOLD signal changes ($\Delta$BOLD) were derived.

## Part 1

**Correlation analysis between cerebral hemodynamic responses and RGE metrics.** The cerebral hemodynamic responses ($\Delta$CBFv and $\Delta$BOLD) were separately correlated with the RGE metrics including bER, $\Delta PCO_2$, $\Delta PO_2$ and ToB. The correlation indicated by Pearson's correlation coefficient was considered significant at $p < 0.05$. Fisher Z-transformation was used to transform Pearson's correlation coefficients to Fisher Z scores for group analysis. Paired t-tests were used to compare the Fisher Z scores representing the correlation between cerebral hemodynamic responses and bER with those indicating the correlation between cerebral hemodynamic responses and RGE metrics other than bER. Differences were considered to be significant at $p < 0.05$.

**Dynamic analysis of coherence between cerebral hemodynamic responses and RGE metrics as function of time and frequency.** Wavelet transform coherence is a method for

analyzing the coherence and phase lag between two time series as a function of both time and frequency [45–47]. It is therefore well suited to investigating non-stationary changes in coupling between the time series of cerebral hemodynamic responses (ΔCBFv and ΔBOLD) and the time series of RGE metrics including bER, $\Delta PCO_2$, $\Delta PO_2$ and ToB, as well as the phase lag of the cerebral hemodynamic responses to each of the metrics. The time series of ΔBOLD were extracted from left gray matter (LGM), right gray matter (RGM), left white matter (LWM) and right white matter (RWM). We used the Matlab wavelet cross-spectrum toolbox developed by Grinsted et al. [46]. Squared wavelet coherence between time series of RGE metrics and cerebral hemodynamic responses (ΔCBFv and ΔBOLD) was separately plotted with x-axis as time and y-axis as scale which had been converted to its equivalent Fourier period. An example of squared wavelet coherence between bER and ΔCBFv in right MCA from a representative subject under breath hold challenge is shown in S2 Fig. The magnitude of wavelet transform coherence ranged between 0 and 1 that can be conceptualized as a localized correlation coefficient in time and frequency space [46]. The phase angle between the two time series at particular samples of the time-frequency plane is indicated by an arrow: a rightward pointing arrow indicates that the time series are in phase, or positively correlation ($\phi = 0$); a leftward pointing arrow indicates anticorrelation ($\phi = \pi$), and the downward and upward pointing arrows indicate phase angles of $\pi/2$ and $-\pi/2$ relative to $\phi = 0$, respectively. Areas inside the 'cone of influence', which are locations in the time-frequency plane where edge effects give rise to lower confidence in the computed values, are shown in faded color outside of the conical contour. The statistical significance level of the wavelet coherence is estimated using Monte Carlo method and the 5% significance level against red noise is shown as thick contour in the squared wavelet coherence plot. The wavelet coherence magnitudes and phases bounded by thick contour outside the cone of influence are considered significant.

Time-averaged coherence is defined as the total significant coherence at each scale of Fourier periods (converted into frequency) where the wavelet coherence magnitude exceeds 95% significance level, normalized by the maximum possible coherence outside the cone of influence at that particular scale (S2 Fig). It is interpreted in the similar way as the coherence in the transfer function analysis which has been used in cerebral autoregulation study [48]. The mean time-averaged coherence at the phase lags of $0\pm\pi/2$ and $\pi\pm\pi/2$ were plotted for all subjects included in TCD or MRI sessions to explore the Fourier periods/frequency bandwidths that oscillations of cerebral hemodynamic responses (ΔCBFv and ΔBOLD) were in synchrony with each physiological time series of bER, $\Delta PO_2$, $\Delta PCO_2$ and ToB when the subjects were performing breath hold task.

## Part 2

**Linear regression for CVR quantification.** Linear regression analysis was used to derive CVR from the time series of ΔBOLD and vasoactive stimulus when the subject was under breath hold and exogenous $CO_2$ challenges. The time series of each of the commonly used vasoactive stimuli (bER, $P_{ET}CO_2$ and ToB) served as a regressor in a separate linear regression analysis. CVR was defined as the percent BOLD signal changes per unit change of the vasoactive stimulus. Therefore CVR was quantified by the coefficient of regression, i.e. the slope.

The statistical parametric maps for individual subjects were cluster-corrected using a threshold estimated with Monte Carlo simulation algorithm. Individual subject brain volume with CVR magnitude was registered onto each subject's anatomical scan and transformed to the standardized space of Talairach and Tournoux [49]. In order to protect against type I error, individual voxel probability threshold of $p<0.005$ was held to correct the overall significance level to $\alpha<0.05$. Monte Carlo simulation was used to correct for multiple comparisons

[50]. Based upon a Monte Carlo simulation with 2000 iteration processed with ClustSim program [51], it was estimated that a $476mm^3$ contiguous volume would provide the significance level $\alpha < 0.05$, which met the overall corrected threshold of $p < 0.05$.

**Group comparison of CVR between breath hold and exogenous $CO_2$ challenges.** For each subject who participated in both breath hold and exogenous $CO_2$ MRI scanning, CVR values were derived from regressing $\Delta BOLD$ on bER ($CVR_{BH-bER}$), $\Delta BOLD$ on $P_{ET}CO_2$ ($CVR_{BH-PETCO2}$) and $\Delta BOLD$ on ToB ($CVR_{BH-ToB}$) when the subjects were under breath hold challenge. $CVR_{BH-bER}$, $CVR_{BH-PETCO2}$ and $CVR_{BH-ToB}$ values were separately averaged in each of the 160 brain regions parcellated by the software FreeSurfer. Using the same method, CVR values during exogenous $CO_2$ challenge were obtained by regressing $\Delta BOLD$ on $P_{ET}CO_2$ ($CVR_{CO2-PETCO2}$). $CVR_{CO2-PETCO2}$ values were averaged in each of the 160 brain regions. To study the CVR changes in group, one-sample t-tests were applied onto the brain volumes with regional $CVR_{BH-bER}$, $CVR_{BH-PETCO2}$, $CVR_{BH-ToB}$ and $CVR_{CO2-PETCO2}$. Differences were considered significant at false discovery rate adjusted $p_{fdr} < 0.05$.

The physiological mechanisms underlying breath holding and exogenous $CO_2$ hypercapnia are potentially different. To study the precision of vasoactive stimulus in predicting the regional cerebral hemodynamic responses, percentage of voxels in each region having significant CVR changes that survived at cluster-corrected $p < 0.05$ in individual subject analysis, in short vCVR, were calculated. Individual subject brain volumes with regional vCVR due to $P_{ET}CO_2$, ($vCVR_{BH-PETCO2}$), ToB ($vCVR_{BH-ToB}$) and bER ($vCVR_{BH-bER}$) were obtained in breath hold MRI scanning, while those with regional vCVR due to $P_{ET}CO_2$ ($vCVR_{CO2-PETCO2}$) were obtained in exogenous $CO_2$ MRI scanning. One sample t-tests were again applied for group analysis. Differences were considered significant at $p_{fdr} < 0.05$.

To study the usefulness of $P_{ET}CO_2$, ToB and bER in CVR quantification under breath hold challenge, using $vCVR_{CO2-PETCO2}$ as reference, paired t-tests were applied to compare the regional brain maps of $vCVR_{CO2-PETCO2}$ under exogenous $CO_2$ challenge with those of $vCVR_{BH-PETCO2}$, $vCVR_{BH-ToB}$ and $vCVR_{BH-bER}$ in groups. Differences were considered significant at $p_{fdr} < 0.05$.

## Results

### Part 1

Subject demographics were shown in Table 1. The $P_{ET}CO_2$, $P_{ET}O_2$, $\Delta PCO_2$, $\Delta PO_2$ and bER measured in the baseline periods, i.e. the period before the first breath hold epoch, in the TCD and MRI sessions were summarized in Tables 2 and 3 respectively. The breath hold duration, the changes of $\Delta PCO_2$, $\Delta PO_2$, bER, $\Delta CBFv$ and $\Delta BOLD$ averaged over the 6 breath hold epochs from the onset to the end of each breath hold epoch were also included. Most of the subjects were able to hold their breaths for 30 seconds on average. The averaged changes of $\Delta PO_2$ from the onset to the end of the breath hold epochs were almost 3 to 4 folds larger than those of $\Delta PCO_2$.

**Correlation among RGE metrics.** The correlations among the RGE metrics (bER, $\Delta PCO_2$, $\Delta PO_2$, $P_{ET}CO_2$, $P_{ET}O_2$, and ToB) in TCD and MRI sessions are shown in S1 Fig. Strong and positive correlation was found between bER and $\Delta PO_2$ consistently in all subjects (Pearson's r, 0.70–0.98, $p < 0.001$), while the correlation between bER and $\Delta PCO_2$ varied from weak to moderate (Pearson's r, 0.07–0.75) (S1 Table). Moderate to strong correlation was observed between $\Delta PO_2$ and $\Delta PCO_2$ (Pearson's r, 0.44–0.88), and between $P_{ET}O_2$ and $P_{ET}CO_2$ (Pearson's r, 0.46–0.91). Such correlations suggest that $\Delta PO_2$ and $\Delta PCO_2$ (or $P_{ET}O_2$ and $P_{ET}CO_2$) are not necessarily redundant. Similar observation of non-redundancy between $P_{ET}O_2$ and $P_{ET}CO_2$ (correlation coefficients, 0.25–0.93) during spontaneous breathing was

**Table 1. Subject demographics and their participation in breath hold and exogenous $CO_2$ challenges in the TCD and MRI sessions.**

| Subjects | Gender | Age | TCD | MRI | |
| --- | --- | --- | --- | --- | --- |
| | | | Breath Hold Challenge | Breath Hold Challenge | Exogenous CO2 Challenge |
| s1 | M | 35 | - | √ | √ |
| s2 | M | 48 | - | √ | √ |
| s3 | M | 22 | - | √ | √ |
| s4 | M | 38 | √ | √ | √ |
| s5 | M | 28 | √ | √ | - |
| s6 | M | 27 | √ | √ | √ |
| s7 | M | 32 | √ | √ | √ |
| s8 | M | 22 | √ | √ | √ |
| s9 | M | 32 | √ | √ | √ |
| s10 | F | 26 | √ | √ | √ |
| s11 | F | 27 | √ | √ | √ |
| s12 | F | 47 | √ | √ | - |
| s13 | M | 46 | - | √ | - |
| s14 | F | 25 | √ | √ | - |
| s15 | F | 23 | √ | √ | - |
| s16 | M | 29 | - | √ | - |
| s17 | F | 23 | √ | - | - |
| Number of subjects: | | | 12 | 16 | 10 |

also reported by Lenfant et al. [30]. The different ranges of correlation strength observed between $\Delta PO_2$ and $\Delta PCO_2$ in the TCD and MRI sessions may be due to the interaction of the subjects with the environment. Subjects were sitting in an open and quiet environment in the TCD sessions while they were in supine position in a noisy MRI scanner bore. Many previous studies reported that a change from supine posture to sitting upright was associated with a redistribution of both blood flow and ventilation in the lungs, which affected the arterial $PO_2$ [52–55]. While it is interesting to observe a difference in the interaction between subjects and environment, the details of such mechanisms are outside the scope of our current study.

**Correlation between cerebral hemodynamic responses and RGE metrics.** Fig 2A shows the time series of $\Delta CBFv$ in left MCA and physiological changes including bER, $\Delta PO_2$, $\Delta PCO_2$, ToB, $P_{ET}O_2$ and $P_{ET}CO_2$ in a representative subject under breath hold challenge, while Fig 2B shows the time series of $\Delta BOLD$ in left gray matter, and the corresponding changes in RGE metrics in the same representative subject. The time series of bER followed the $\Delta CBFv$ and $\Delta BOLD$ closely in the subject. By visual inspection of the time series of $\Delta CBFv$ and $\Delta BOLD$, the interpolated values of $P_{ET}CO_2$ and $\Delta PCO_2$ during breath hold periods did not necessarily follow the changes of $\Delta CBFv$ and $\Delta BOLD$.

Among the four RGE metrics of bER, $\Delta PO_2$, $\Delta PCO_2$ and ToB, bER is the only parameter that consistently showed strong correlation with the $\Delta CBFv$ measured in left and right MCAs (Pearson's r, 0.40–0.89, p<0.001) (Fig 3A). The bER also correlated with $\Delta BOLD$ extracted from gray (LGM and RGM) and white matter (LWM and RWM) under breath hold challenge (Pearson's r, 0.21–0.83, p<0.05), except one subject in the left brain (Pearson's r, -0.011–0.18, p>0.05) (Fig 3A). The results of correlation analyses between cerebral hemodynamic responses and RGE metrics were summarized in S2 Table. Group comparisons of the correlation between cerebral hemodynamic responses ($\Delta CBFv$ and $\Delta BOLD$) and RGE metrics (bER, $\Delta PO_2$, $\Delta PCO_2$ and ToB) also showed that the correlation between cerebral hemodynamic responses and bER was significantly stronger than those between cerebral hemodynamic

**Table 2. RGE metrics and ΔCBFv in TCD sessions.**

| Subjects | Baseline | | | | | Averaged Changes From the Onset to the End of Breath Hold Epochs | | | | | |
|---|---|---|---|---|---|---|---|---|---|---|---|
| | $P_{ET}CO_2$ (mmHg) | $P_{ET}O_2$ (mmHg) | $\Delta PCO_2$ (mmHg) | $\Delta PO_2$ (mmHg) | bER | Breath Hold Duration (seconds) | $\Delta PCO_2$ (mmHg) | $\Delta PO_2$ (mmHg) | bER | ΔCBFv in LMCA (%) | ΔCBFv in RMCA (%) |
| s4 | 39.6 (0.5) | 108.5 (0.6) | 39.7 (0.5) | 42.7 (0.6) | 1.1 (0.0) | 30.7 (1.3) | 8.9 (7.7) | 25.1 (14.2) | 0.4 (0.1) | 43.4 (9.2) | 43.4 (9.1) |
| s5 | 39.5 (1.7) | 112.0 (2.5) | 37.3 (4.4) | 44.6 (5.4) | 1.2 (0.0) | 35.5 (0.6) | 3.4 (0.8) | 18.2 (2.4) | 0.4 (0.1) | 21.1 (4.6) | 31.8 (2.7) |
| s6 | 34.3 (0.5) | 116.5 (0.7) | 33.6 (0.5) | 32.2 (0.7) | 1.0 (0.0) | 33.1 (1.4) | 7.7 (4.1) | 32.2 (9.5) | 0.6 (0.1) | 37.2 (6.5) | 40.5 (10.3) |
| s7 | 39.2 (1.0) | 102.9 (1.5) | 39.3 (1.0) | 48.7 (1.7) | 1.2 (0.0) | 31.7 (2.4) | 8.3 (1.5) | 38.6 (5.7) | 0.7 (0.1) | 51.3 (12.6) | 46.7 (11.2) |
| s8 | 38.3 (0.7) | 104.6 (1.2) | 37.3 (0.6) | 46.4 (1.4) | 1.2 (0.0) | 33.0 (1.4) | 5.6 (3.7) | 27.4 (11.4) | 0.5 (0.1) | 42.5 (7.9) | 41.9 (7.4) |
| s9 | 34.7 (0.4) | 106.4 (0.9) | 35.1 (0.4) | 45.9 (1.0) | 1.3 (0.0) | 38.2 (4.4) | 2.8 (3.6) | 11.4 (17.8) | 0.2 (0.3) | 28.7 (12.9) | 28.6 (8.8) |
| s10 | 30.7 (0.8) | 116.2 (2.2) | 30.6 (0.4) | 36.8 (1.8) | 1.2 (0.0) | 34.1 (1.4) | 8.5 (1.0) | 39.3 (6.0) | 0.8 (0.2) | 31.1 (9.0) | 32.6 (7.6) |
| s11 | 35.6 (0.4) | 109.9 (0.7) | 35.0 (0.4) | 40.3 (0.8) | 1.2 (0.0) | 32.6 (1.5) | 4.7 (1.5) | 27.7 (3.1) | 0.6 (0.1) | — | 39.5 (7.4) |
| s12 | 35.3 (0.2) | 116.0 (0.5) | 34.6 (0.2) | 35.0 (0.8) | 1.0 (0.0) | 32.1 (1.5) | 8.7 (1.3) | 40.3 (3.0) | 0.8 (0.0) | 43.6 (2.3) | 52.5 (1.7) |
| s14 | 33.9 (0.3) | 121.2 (0.7) | 30.1 (0.5) | 32.0 (1.1) | 1.1 (0.0) | 31.7 (3.8) | 3.9 (1.6) | 17.5 (2.3) | 0.5 (0.1) | 51.1 (3.2) | 46.1 (5.8) |
| s15 | 31.9 (0.8) | 118.8 (0.7) | 28.3 (1.3) | 32.8 (1.2) | 1.2 (0.0) | 30.6 (2.3) | 8.0 (3.9) | 27.5 (3.5) | 0.7 (0.2) | 53.0 (7.0) | 50.8 (7.0) |
| s17 | 31.9 (2.1) | 120.2 (2.8) | 27.5 (3.5) | 31.3 (4.5) | 1.1 (0.0) | 23.3 (2.7) | 2.5 (3.7) | 15.8 (9.8) | 0.5 (0.3) | 38.9 (9.1) | 36.1 (5.7) |

Baseline mean values (SD) of $P_{ET}CO_2$, $P_{ET}O_2$, $\Delta PCO_2$, $\Delta PO_2$ and bER for all subjects (n = 12) who were under breath hold challenge in the TCD sessions (*left*). The averaged changes (SD) of breath hold duration, $\Delta PCO_2$, $\Delta PO_2$, bER, ΔCBFv measured in LMCA and RMCA from the onset to the end of the breath hold epochs in the TCD sessions (*right*).

responses and the RGE metrics other than bER (S2 Table). Although ToB was not as good as bER in correlating with ΔCBFv or ΔBOLD, it was better than $\Delta PCO_2$ to serve as an indicator of the breath hold periods.

**Dynamic analysis of coherence between cerebral hemodynamic responses and RGE metrics as function of time and frequency.** The mean time-averaged coherence between time series of RGE metrics (bER, $\Delta PO_2$, $\Delta PCO_2$ and ToB) and cerebral hemodynamic responses (ΔCBFv and ΔBOLD) was found to be significantly stronger between 0.008Hz (1/128 seconds) and 0.03Hz (1/32 seconds) at phase lag of 0±π/2 (S3 Fig). We therefore focused on the distribution of time-averaged coherence between RGE metrics and cerebral hemodynamic responses at the phase lag of 0±π/2 (Fig 3). Among the 4 RGE metrics, the mean time-averaged coherence between bER and cerebral hemodynamic responses at phase lag of 0±π/2 was the strongest at the frequency bandwidths of 0.008–0.03 Hz while that between $\Delta PCO_2$ and cerebral hemodynamic responses was the weakest at the same frequency bandwidths. The mean time-averaged coherence between bER, $\Delta PO_2$, ToB and cerebral hemodynamic responses at phase lag of 0±π/2 reached 0.6 or above. at the frequency bandwidths of 0.008–0.03 Hz. The correlation and dynamic coherence results suggest that CBFv and BOLD signals oscillated with bER at a broad frequency range of low frequencies when the subjects were performing breath hold task. The differences in the correlation and coherence findings between $\Delta PO_2$ and $\Delta PCO_2$ further suggest that changes of $PO_2$ and $PCO_2$ are not simply the inverse of each other.

**Table 3. RGE metrics and ΔBOLD in MRI sessions.**

| Subjects | Baseline | | | | | Averaged Changes From the Onset to the End of Breath Hold Epochs | | | | | | | |
|---|---|---|---|---|---|---|---|---|---|---|---|---|---|
| | $P_{ET}CO_2$ (mmHg) | $P_{ET}O_2$ (mmHg) | $\Delta PCO_2$ (mmHg) | $\Delta PO_2$ (mmHg) | bER | Breath Hold Duration (seconds) | $\Delta PCO_2$ (mmHg) | $\Delta PO_2$ (mmHg) | bER | ΔBOLD in LGM (%) | ΔBOLD in RGM (%) | ΔBOLD in LWM (%) | ΔBOLD in RWM (%) |
| s1 | 42.9 (0.7) | 99.3 (2.3) | 42.5 (0.7) | 54.0 (2.8) | 1.3 (0.0) | 33.4 (3.1) | 3.9 (2.4) | 29.7 (7.6) | 0.6 (0.1) | 2.3 (0.6) | 2.2 (0.4) | 1.0 (0.2) | 1.0 (0.2) |
| s2 | 38.7 (0.1) | 113.8 (0.2) | 36.6 (0.6) | 32.7 (0.4) | 0.9 (0.0) | 31.7 (1.4) | 9.5 (2.7) | 40.5 (2.5) | 0.8 (0.1) | 2.1 (0.7) | 2.2 (0.5) | 1.1 (0.2) | 1.2 (0.2) |
| s3 | 36.1 (0.3) | 112.7 (1.1) | 35.9 (0.3) | 38.1 (1.7) | 1.1 (0.0) | 32.6 (2.7) | 5.3 (1.9) | 46.8 (5.3) | 1.0 (0.2) | 2.3 (1.3) | 2.5 (1.6) | 1.3 (0.6) | 1.3 (0.8) |
| s4 | 39.4 (0.7) | 104.1 (1.2) | 39.2 (0.7) | 44.2 (1.5) | 1.1 (0.0) | 35.5 (1.4) | 1.7 (1.3) | 17.3 (6.8) | 0.4 (0.1) | 2.5 (0.4) | 3.7 (1.1) | 1.3 (0.2) | 1.6 (0.6) |
| s5 | 32.4 (0.4) | 117.1 (0.4) | 32.1 (0.4) | 32.4 (0.5) | 1.0 (0.0) | 35.3 (1.3) | 7.6 (5.4) | 30.9 (6.0) | 0.7 (0.1) | 1.6 (0.6) | 1.5 (0.3) | 0.7 (0.2) | 0.9 (0.4) |
| s6 | 39.0 (0.6) | 109.8 (0.6) | 37.5 (0.8) | 38.3 (0.9) | 1.0 (0.0) | 32.2 (2.1) | 9.4 (3.9) | 39.9 (7.7) | 0.8 (0.1) | 4.0 (1.2) | 3.5 (1.2) | 2.1 (0.4) | 1.6 (0.6) |
| s7 | 40.3 (0.6) | 109.0 (2.4) | 38.6 (0.5) | 41.3 (2.8) | 1.1 (0.1) | 32.9 (1.5) | 3.5 (3.5) | 34.3 (6.4) | 0.7 (0.1) | 1.9 (0.1) | 2.2 (0.5) | 1.2 (0.7) | 0.9 (0.2) |
| s8 | 37.7 (0.2) | 103.1 (0.3) | 37.1 (0.2) | 47.9 (0.4) | 1.3 (0.0) | 35.4 (1.1) | 3.0 (3.7) | 22.4 (14.9) | 0.6 (0.0) | 2.7 (0.6) | 2.9 (0.8) | 1.3 (0.4) | 1.2 (0.3) |
| s9 | 41.3 (0.4) | 106.6 (1.4) | 40.0 (0.4) | 44.6 (1.7) | 1.1 (0.0) | 33.9 (4.7) | 6.8 (4.0) | 35.9 (21.3) | 0.7 (0.4) | 1.9 (0.6) | 2.1 (0.7) | 1.0 (0.4) | 0.9 (0.2) |
| s10 | 36.6 (0.2) | 117.6 (0.9) | 34.3 (0.2) | 35.9 (0.9) | 1.0 (0.0) | 31.5 (1.2) | 8.3 (0.9) | 50.9 (4.1) | 1.0 (0.1) | 4.3 (1.6) | 2.5 (0.4) | 2.0 (0.8) | 1.2 (0.2) |
| s11 | 34.2 (0.2) | 114.7 (0.7) | 33.1 (0.2) | 35.7 (0.9) | 1.1 (0.0) | 36.4 (1.7) | 0.0 (0.6) | 4.5 (5.5) | 0.2 (0.1) | 2.0 (0.4) | 1.6 (0.1) | 1.2 (0.4) | 0.9 (0.4) |
| s12 | 37.6 (0.4) | 113.6 (1.5) | 36.0 (0.4) | 40.0 (1.7) | 1.1 (0.0) | 32.0 (1.1) | 1.6 (1.6) | 26.0 (4.9) | 0.8 (0.1) | 2.6 (0.4) | 2.7 (0.6) | 1.6 (0.4) | 1.3 (0.4) |
| s13 | 36.3 (0.6) | 109.1 (3.7) | 34.1 (0.5) | 44.4 (4.5) | 1.3 (0.1) | 31.8 (4.1) | 3.7 (6.3) | 15.8 (16.0) | 0.4 (0.2) | 1.9 (0.2) | 2.0 (0.3) | 1.0 (0.1) | 1.0 (0.1) |
| s14 | 36.1 (0.9) | 114.6 (0.9) | 31.7 (0.6) | 38.1 (0.8) | 1.2 (0.0) | 35.7 (0.8) | 3.6 (1.6) | 17.2 (4.4) | 0.4 (0.1) | 2.0 (0.9) | 1.5 (0.3) | 0.9 (0.1) | 0.8 (0.2) |
| s15 | 35.8 (0.5) | 115.9 (0.9) | 32.8 (0.9) | 37.7 (1.3) | 1.2 (0.0) | 34.1 (0.9) | 7.9 (5.3) | 30.7 (8.4) | 0.6 (0.1) | 1.4 (0.2) | 4.3 (4.1) | 0.9 (0.1) | 1.6 (0.9) |
| s16 | 34.8 (0.5) | 119.9 (1.0) | 31.4 (0.5) | 23.9 (1.2) | 0.8 (0.0) | 27.7 (6.5) | 1.6 (2.1) | 11.8 (4.0) | 0.3 (0.1) | 1.3 (0.6) | 1.3 (0.4) | 0.7 (0.4) | 0.6 (0.2) |

Baseline mean values (SD) of $P_{ET}CO_2$, $P_{ET}O_2$, $\Delta PCO_2$, $\Delta PO_2$ and bER for all subjects (n = 16) who were under breath hold challenge in the MRI sessions (*left*). The averaged changes (SD) of breath hold duration, $\Delta PCO_2$, $\Delta PO_2$, bER, ΔBOLD measured in LGM, RGM, LWM and RWM from the onset to the end of the breath hold epochs in the MRI sessions (*right*).

## Part 2

**Group comparison of CVR between breath hold and exogenous $CO_2$ challenges.** Fig 4 shows the regional CVR brain maps averaged across 16 subjects who performed breath hold task in MRI sessions. Under breath hold challenge, most of the brain regions showed significant increase in $CVR_{BH-bER}$ and $CVR_{BH-ToB}$ especially in thalamus, insula and putamen, while no significant changes of $CVR_{BH-PETCO2}$ were observed in the same subject group (Fig 4).

In the comparison of CVR brain maps under breath hold and exogenous $CO_2$ challenge, a subset of 10 subjects who participated in both challenges were included. Under exogenous $CO_2$ challenge, most of the brain regions showed increased $CVR_{CO2-PETCO2}$ in the subject group especially thalamus, insula and putamen (Fig 5A), and the $vCVR_{CO2-PETCO2}$ which had significant $CVR_{CO2-PETCO2}$ changes exceeded 80% in most of the brain regions (Fig 5B). For

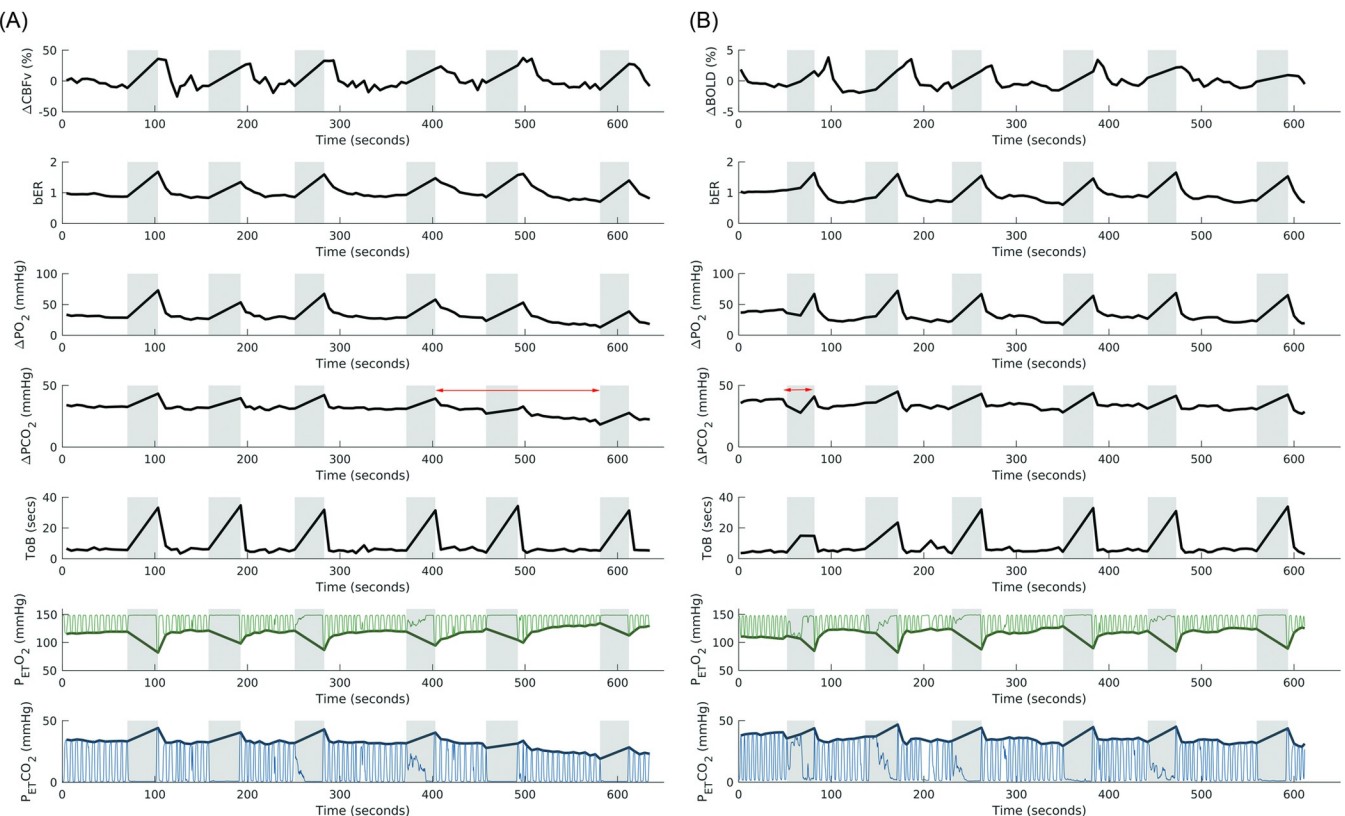

**Fig 2. Time series of cerebral hemodynamic responses and RGE metrics in a representative subject under breath hold challenge.** (A) Time series of ΔCBFv in left MCA and physiological changes including breath-by-breath bER, $\Delta PO_2$, $\Delta PCO_2$, ToB, $P_{ET}O_2$ and $P_{ET}CO_2$ in a representative subject under breath hold challenge in TCD session. (B) Time series of ΔBOLD in left gray matter (LGM) and the corresponding physiological changes in the same representative subject under breath hold challenge in MRI session. Shaded areas represent breath hold periods. The time series of bER followed closely to the ΔCBFv and ΔBOLD changes, while $\Delta PCO_2$ did not follow ΔCBFv and ΔBOLD changes some time during the challenges as indicated by two-headed arrows in red. Thin green lines represent partial pressure of $O_2$, while the thin blue lines represent partial pressure of $CO_2$. Thick green lines and thick blue lines represent $P_{ET}O_2$ and $P_{ET}CO_2$ respectively.

the same group of subjects under breath hold challenge, increased $CVR_{BH\text{-}bER}$ and $CVR_{BH\text{-}ToB}$ were found in most of the brain regions, while no significant changes of $CVR_{BH\text{-}PETCO2}$ were shown in most of the brain regions (Fig 5A). CVR brain maps under breath hold challenge shown in Fig 5A (n = 10) were consistent with those shown in Fig 4 (n = 16). Comparing with the $vCVR_{BH\text{-}ToB}$ and $vCVR_{BH\text{-}PETCO2}$, $vCVR_{BH\text{-}bER}$ showed the largest percentage of voxels with significant CVR changes in different brain regions, implying a significantly high precision of bER predicting regional ΔBOLD in breath hold challenge (Fig 5B). The paired comparison between $vCVR_{CO2\text{-}PETCO2}$ and $vCVR_{BH\text{-}bER}$, as well as that between $vCVR_{CO2\text{-}PETCO2}$ and $vCVR_{BH\text{-}ToB}$, did not show significant difference in most of the brain regions, while significant differences were found in most of the brain regions between $vCVR_{CO2\text{-}PETCO2}$ and $vCVR_{BH\text{-}PETCO2}$ (Fig 5C). Such findings suggest that bER in breath hold challenge is more appropriate to be used as vasoactive stimulus than $P_{ET}CO_2$ in assessing regional CVR under breath hold challenge.

## Discussion

Our findings show a strong positive correlation between the cerebral hemodynamic responses and our new breath-by-breath $O_2$-$CO_2$ exchange ratio, in short bER, under brief breath hold

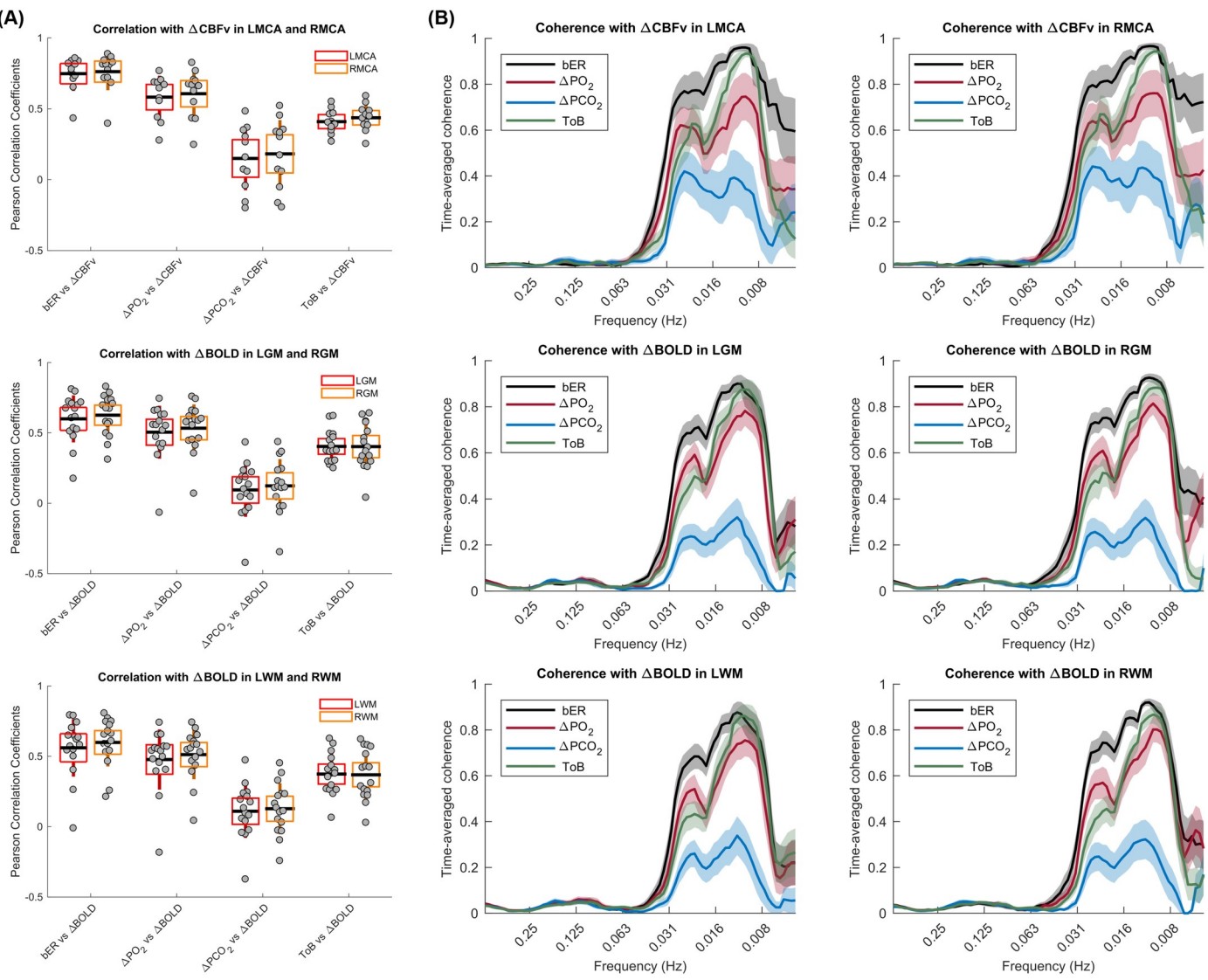

**Fig 3. Correlation and coherence between cerebral hemodynamic responses and RGE metrics in all subjects under breath hold challenge.** (A) Strength of correlation between cerebral hemodynamic responses and respiratory metrics including bER, $\Delta PO_2$, $\Delta PCO_2$ and ToB. Top panel shows the correlation with $\Delta CBFv$ in LMCA and RMCA (n = 12), the middle panel shows the correlation with $\Delta BOLD$ in LGM and RGM (n = 16), and the bottom panel shows the correlation with $\Delta BOLD$ in LWM and RWM (n = 16). Each grey circle represents the Pearson's correlation coefficient from the correlation analysis of the parameter pairs shown on x-axis for each subject. The thick horizontal black line, the box and the vertical rod represent the mean, 95% confidence interval and standard deviation of the group data respectively. The cerebral hemodynamic responses correlate with the respiratory metrics consistently in a descending order of bER, $\Delta PO_2$, ToB and $\Delta PCO_2$. (B) Distribution of time-averaged coherence at the phase lag $0\pm\pi/2$ between time series of respiratory metrics (bER, $\Delta PO_2$, $\Delta PCO_2$ and ToB) and cerebral hemodynamic responses. Top panel shows the coherence with $\Delta CBFv$ in LMCA and RMCA (n = 12), the middle panel shows the coherence with $\Delta BOLD$ in LGM and RGM (n = 16), and the bottom panel shows the coherence with $\Delta BOLD$ in LWM and RWM (n = 16). The mean time-averaged coherence in the frequency bandwidths from 0.008 to 0.25Hz were plotted (thick color lines). Color shaded areas represent standard error of the mean. The mean time-averaged coherence between bER and cerebral hemodynamic responses reached 0.6 or above at the frequency range from 0.008Hz (1/128 seconds) to 0.03Hz (1/32 seconds), while the mean time-averaged coherence between $\Delta PCO_2$ and cerebral hemodynamic responses stayed below 0.4. Comparing with $\Delta PCO_2$, the mean time-averaged coherence of $\Delta PO_2$ with cerebral hemodynamic responses reached 0.5 or above in the frequency range of 0.008–0.03Hz, which was better than that of $\Delta PCO_2$.

challenge. We are the first to show that the dynamic changes in bER robustly characterized CBFv and BOLD responses much better than changes in $P_{ET}CO_2$ or ToB under breath hold challenge in the very low frequency range of 0.008–0.03Hz. The difference between bER and $\Delta PCO_2$ in coherence with cerebral hemodynamic responses within the frequency range of 0.008–0.03 Hz cannot be attributed to the long periods of the breath hold protocol alone since

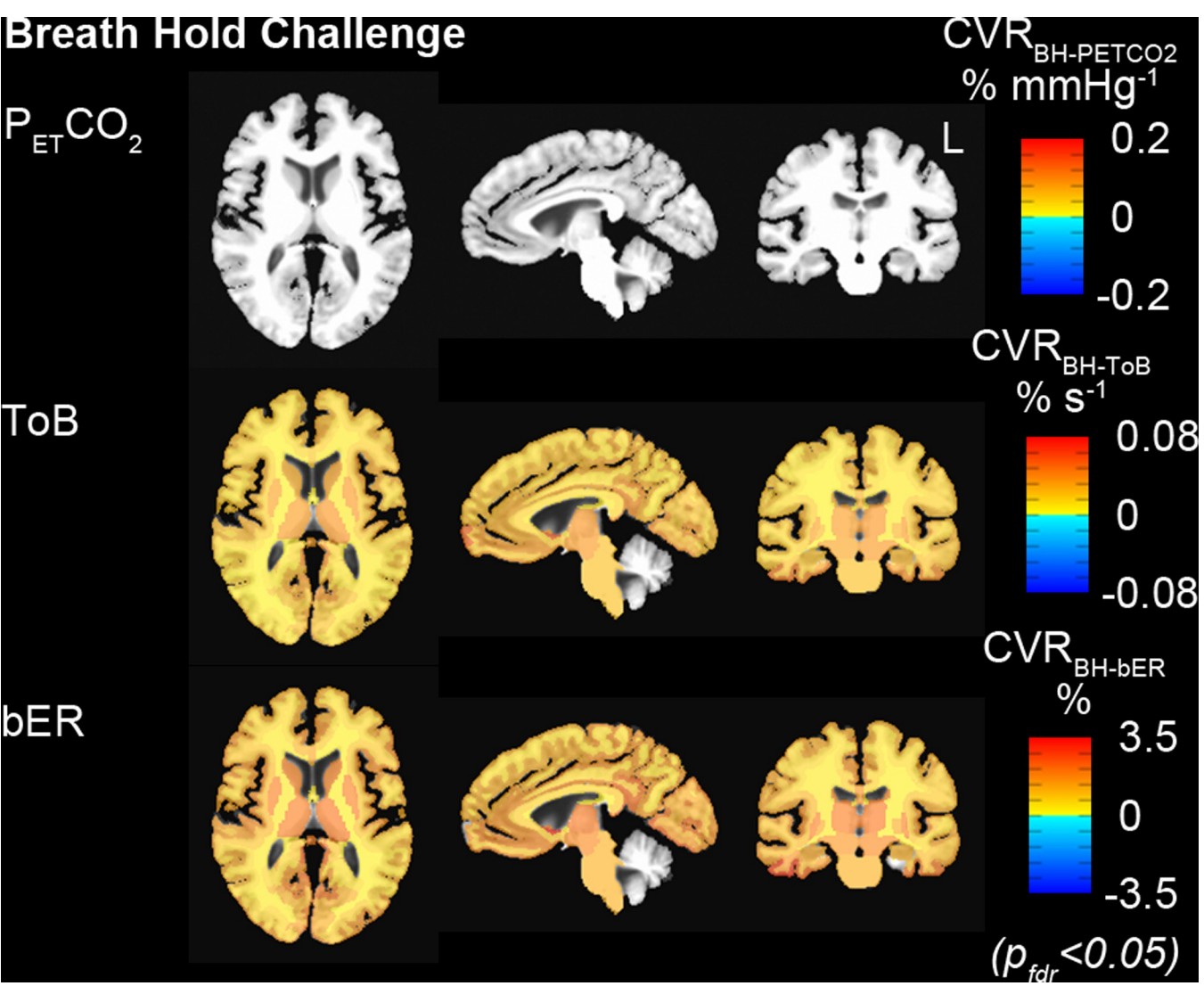

**Fig 4. Group CVR maps (CVR$_{BH-PETCO2}$, CVR$_{BH-ToB}$ and CVR$_{BH-bER}$) generated by regressing ΔBOLD separately on P$_{ET}$CO$_2$, ToB and bER under breath hold challenges (n = 16).** All the CVR maps had been corrected for p$_{fdr}$<0.05. Maps of CVR$_{BH-ToB}$ and CVR$_{BH-bER}$ were comparable, while P$_{ET}$CO$_2$ was not able to characterize the regional ΔBOLD.

the influence of the periods of the breath hold protocol should contribute equally to all RGE metrics. During breath holding, we presented the combined effect of both hypoxia and hypercapnia on the cerebral hemodynamic responses measured using TCD and BOLD-fMRI. Given that the concurrent changes of P$_{ET}$O$_2$ and P$_{ET}$CO$_2$ are in opposite direction and the magnitudes depend on the respiratory phase and volume, ΔPO$_2$ and ΔPCO$_2$ were in phase and selected here to more conveniently characterize the breath-by-breath changes during breath holding and spontaneous breathing epochs. bER was selected with the O$_2$ term as the numerator of the O$_2$/CO$_2$ ratio due to bER's positive temporal relationship with CBF changes. In characterizing regional BOLD signal changes to brief breath hold challenge, bER which took into account the interaction of ΔPO$_2$ and ΔPCO$_2$ yielded much better results than what P$_{ET}$CO$_2$ and ToB could do as we showed that the brain regions outlined by bER during brief breath hold challenge were comparable with those outlined by the P$_{ET}$CO$_2$ during exogenous CO$_2$ challenge.

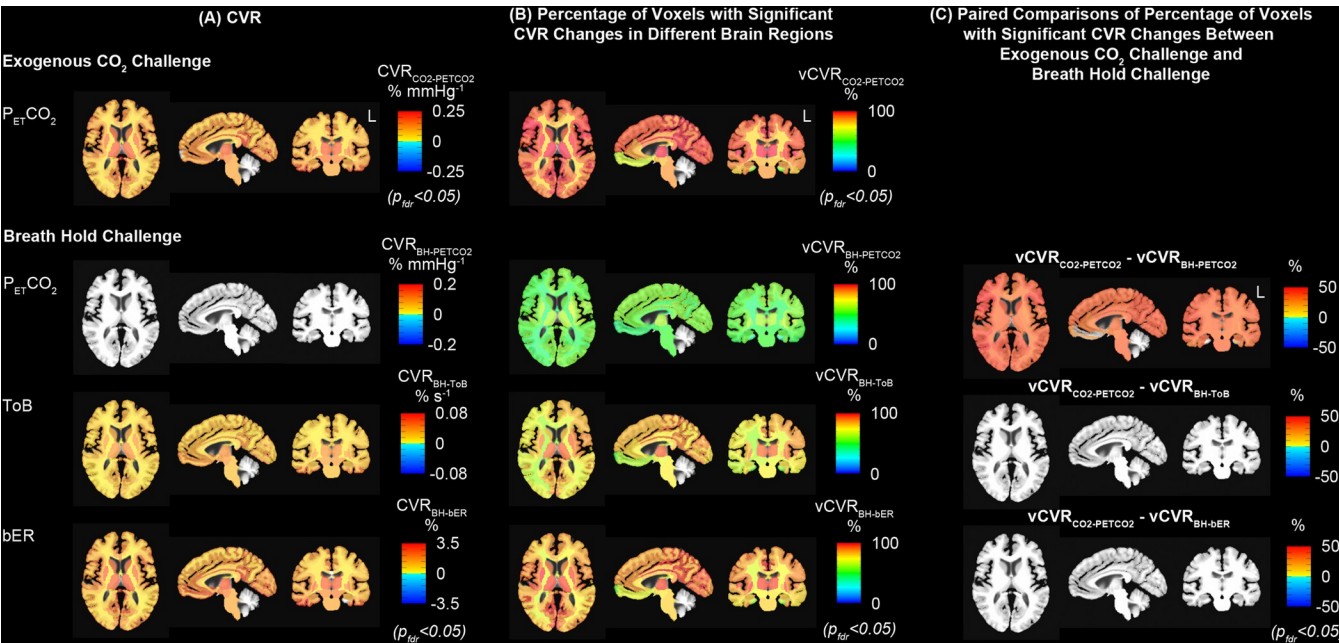

**Fig 5. Comparison of group CVR maps under exogenous $CO_2$ challenge with those under breath hold challenge.** (A) Group CVR showing significant CVR changes under exogenous $CO_2$ and breath hold challenges (n = 10). Group map of $CVR_{BH-PETCO2}$ indicated that ΔBOLD in most of the brain regions did not associate with changes of $P_{ET}CO_2$ under breath hold challenge. (B) Group vCVR maps showing percentage of voxels with significant CVR changes under exogenous CO2 and breath hold challenges for the same group of subjects (n = 10). The percentage of voxels with significant CVR changes under breath hold challenge were found in an increasing order of $vCVR_{BH-PETCO2}$, $vCVR_{BH-ToB}$ and $vCVR_{BH-bER}$. $CVR_{BH-bER}$ and $vCVR_{BH-bER}$ showed large resemblance with $CVR_{CO2-PETCO2}$ and $vCVR_{CO2-PETCO2}$ respectively. (C) Paired comparisons of vCVR maps showed that $vCVR_{CO2-PETCO2}$ maps under exogenous $CO_2$ challenge were not significantly different from $vCVR_{BH-bER}$ maps under breath hold challenge. Group map of $vCVR_{BH-PETCO2}$ showed that ΔBOLD in only small number of voxels within the brain regions associated with the $P_{ET}CO_2$ changes under breath hold challenge, resulting in significant differences found in multiple brain regions in the paired comparison between $vCVR_{CO2-PETCO2}$ and $vCVR_{BH-PETCO2}$.

## Mild hypoxia enhances sensitivity of CBF changes to $CO_2$

During breath holding periods, the $CO_2$ release are dependent on $O_2$ uptake in the closed circuit of systemic circulation created by holding one's breath. The interaction between $ΔPO_2$ and $ΔPCO_2$ during breath hold is mainly resulting from the systemic metabolic process and different from the effect of exogenous gas administration which is primarily indicated by an increase in $ΔPCO_2$. In the current study, $ΔPO_2$ could go from 11 to 51 mmHg and $ΔPCO_2$ could go from 2 to 10 mmHg at the end of 30 seconds of breath holding in the TCD and MRI sessions (Tables 2 and 3). This relatively modest level of change in $ΔPO_2$ had been reported to be able to induce a progressive increase of CBF in the presence of mild hypercapnia [27, 28]. Our findings of $ΔPO_2$ vs. $ΔPCO_2$ support the synergistic effect of hypoxia and hypercapnia on CBF change under breath hold challenge which are in parallel with the increased chemoreceptor activities in the presence of both mild hypoxia and mild hypercapnia reported by several research teams on animal models [15, 17, 18, 56]. Sensitivity of peripheral chemoreceptor at carotid body to arterial $PCO_2$ was found to increase when there was a decrease in the level of arterial $PO_2$ in these studies [15, 17, 18, 56]. The chemoreceptor activity in response to arterial $PO_2$ took place at all levels of $PO_2$ ranging from hypoxia to normoxia and then up to 190mmHg or even higher in the range of hyperoxia. The chemoreceptor response curve to $PO_2$ was similar to a hyperbola with chemoreceptor activity rising faster under hypoxia (below 95mmHg) than under normoxia or hyperoxia [56]. Modulation of peripheral chemoreceptor activities may be expected leading to modulation of CBF while their physiological mechanisms remain to be clarified.

## Wavelet transform coherence (WTC) analysis showed strong coherence between bER and cerebral hemodynamic responses (ΔCBFv and ΔBOLD) under breath hold challenge

We used wavelet transform coherence analysis to examine the temporal features and frequency characteristics of bER, $\Delta PO_2$, $\Delta PCO_2$ and ToB, and their coherence with CBFv and BOLD signal changes under breath hold challenge. Using WTC, we showed that the coherence between cerebral hemodynamic responses (both ΔCBFv and ΔBOLD) and change in bER was much stronger than that between cerebral hemodynamic responses and change in $\Delta PCO_2$ in a wide frequency range of 0.008–0.03Hz (Fig 3). The meaning of the range of 0.008–0.03Hz is best appreciated by examining the WTC findings which showed that bER and $\Delta PCO_2$ were very different in the frequency distribution of their coherence with cerebral hemodynamic responses. The special characteristics of the frequency range of 0.008–0.03Hz need to be considered in the context of each individual RGE metric ($\Delta PO_2$, $\Delta PCO_2$ and bER) separately.

Comparing with $\Delta PCO_2$, the stronger coherence found between bER and cerebral hemodynamic responses in low frequency range of 0.008–0.03Hz may be associated with low frequency physiological processes in the brain that are grouped in B-wave frequency bandwidth. B-waves with a period of 0.5 to 2 minutes have been reported to be related to autoregulation of microvasculature and spontaneous rhythmic oscillations in intracranial pressure [57, 58]. Changes in systemic $PO_2$ and $PCO_2$ may trigger the underlying mechanisms which involve the interaction of central and peripheral respiratory chemoreceptors as well as autonomic system to regulate CBF. B-waves may be associated in such a process through the contractile properties of pericytes or vascular smooth muscle cells to alter vascular diameter and ensure the delivery of $O_2$ and other circulating metabolites [59–62]. As autoregulation of CBF and oxygen delivery are the outcomes in this part of vital homeostatic process [17], it is reasonable that the stronger coupling was found between bER and cerebral hemodynamic responses, and between $\Delta PO_2$ and cerebral hemodynamic responses.

In summary, among the four RGE metrics of bER, $\Delta PO_2$, $\Delta PCO_2$ and ToB, the cerebral hemodynamic responses showed the strongest correlation and dynamic coherence with bER, followed by $\Delta PO_2$, ToB, and $\Delta PCO_2$. A wide range of correlations found between $\Delta PO_2$ and $\Delta PCO_2$ (and between $P_{ET}O_2$ and $P_{ET}CO_2$) in the subject group, as well as the differences in the correlation in the time domain and coherence findings in wavelet transform coherence analysis between $\Delta PO_2$ and $\Delta PCO_2$ indicate that $PO_2$ and $PCO_2$ are not simply inverse of each other. The strong correlation between bER and $\Delta PO_2$ (S1 Fig) indicates that bER is predominantly affected by $\Delta PO_2$. ToB yielded superior correlation result than $\Delta PCO_2$ because ToB indirectly takes into account the duration for both hypoxemia and hypercapnia which both elevate cerebral hemodynamic responses without being affected by the depth of breathing. bER is the most accurate in characterizing cerebral hemodynamic responses under breath hold challenge because it directly takes into account the effect of both $\Delta PO_2$ and $\Delta PCO_2$.

## CVR map from breath hold challenge resembled that from exogenous $CO_2$ challenge

Our CVR findings showed that there was a large resemblance between regional $CVR_{BH-bER}$ and $CVR_{CO2-PETCO2}$, even though the underlying physiological mechanisms for the CBF increase are potentially different (Fig 5). From spontaneous breathing to breath holding periods, the bER, $\Delta PO_2$ and $\Delta PCO_2$ measured were due to endogenous changes of respiratory gases involved in the process of systemic metabolism. Previous studies reported that apnea-induced hypoxia and hypercapnia caused chemoreceptor-mediated central vasodilation and concurrent peripheral vasoconstriction to conserve oxygen delivery to the brain [25], leading

to an increase in CBF and a decrease in peripheral oxygen saturation [26]. Holding breath is different from exogenous $CO_2$ administration where $\Delta PO_2$ and $\Delta PCO_2$ from normocapnic to hypercapnic epochs depended on the externally administered gas mixture and the increase in cerebral hemodynamic responses was mainly due to hypercapnia. One explanation for the large resemblance of the CVR maps between the breath hold and exogenous $CO_2$ challenges is that bER as an optimal regressor under breath hold challenge is able to characterize the increase in cerebral hemodynamic responses in most of the brain regions. Comparing the brain maps of $vCVR_{BH-bER}$ with those of $vCVR_{BH-ToB}$ and $vCVR_{BH-PETCO2}$, more voxels showed significant association between $\Delta BOLD$ and bER. For CVR maps, $CVR_{BH-ToB}$ resembled $CVR_{CO2-PETCO2}$ more than $CVR_{BH-PETCO2}$ did because ToB is a good time indicator of breath hold epochs as shown in Fig 2.

Breath hold challenge, as shown in our findings in Part 2, was able to offer significant regional CVR quantification, as long as the proper regressor was used. Since bER is a measure of the dynamic change of RGE which is related to ratio of the change of partial pressure in $O_2$ uptake to that in $CO_2$ release in the body, mapping CVR to breath hold may offer a novel probe to study the interaction between brain responses and some measures of systemic metabolism. Mild exogenous $CO_2$ challenge is expected to have little effect on cerebral [63] and presumably systemic metabolism. From a technical approach, breath-holding challenge for CVR assessment is much easier to set up in clinical settings than external $CO_2$ administration which requires an elaborate gas administration circuit [6]. The breath-holding challenge also allows patients to switch back to normal breathing whenever their physiological limit is reached [64]. In patients who have compromised vasculature with potential risk of acute intracranial hemorrhage, breath-holding may offer an option for CVR study if one has some yet-to-be proven concern over the more powerful physiological stress presented by rapid increase of externally administered $CO_2$.

## Why was the ratio of $\Delta PO_2$ to $\Delta PCO_2$ used, and not the product?

The success of bER in better characterizing dynamic CBFv and BOLD signal changes under breath hold challenge is closely related to the fact that bER is a ratio which factors out at the same time effects of ventilatory volume fluctuations [32] common to both $\Delta PO_2$ and $\Delta PCO_2$. Given that both hypoxia and hypercapnia induce the increase in cerebral hemodynamic responses, one may wonder whether the product of $\Delta PO_2$ and $\Delta PCO_2$ would be better than bER for the evaluation of change in cerebral hemodynamic responses. Our answer is no. First, the product of $\Delta PO_2$ and $\Delta PCO_2$ would exacerbate the contribution of fluctuations from ventilatory volume. Secondly, unlike the ratio which has long been used to properly describe RGE since at least 1913 [65], the product does not have an established physiological meaning. Actually, another way to look at bER is that it describes the change of $\Delta PO_2$ per unit change of $\Delta PCO_2$, so the ratio provides a way to quantify how $O_2$ and $CO_2$ can work together to interact with cerebral hemodynamic responses.

In a related framework, the 'stimulus index' (SI; $P_{ET}CO_2/P_{ET}O_2$) developed by Bruce et al. [66] for breath hold study is different from bER, namely $\Delta PO_2/\Delta PCO_2$, in its interaction with CBF. Both Bruce et al. (using SI) and our team (using bER) agree upon the influence of $O_2$ and $CO_2$ on CBFv change in breath hold. But there are several major differences between SI and bER.

One major property that contributes to the differences between bER and SI is the direction/phase of oscillations of gas measurements. As shown in Fig 2, the time courses of $P_{ET}O_2$ and $P_{ET}CO_2$ oscillate out of phase, while $\Delta PO_2$ oscillates in phase with $\Delta PCO_2$. With $\Delta PO_2$ being in phase with $\Delta PCO_2$, shared but less interesting physiological signals (e.g. respiratory fluctuation) would be largely factored out in the ratio $\Delta PO_2/\Delta PCO_2$ (bER). With $P_{ET}O_2$ and $P_{ET}CO_2$

being out of phase, those less interesting physiological signals do not get factored out by $P_{ET}CO_2/P_{ET}O_2$ (SI) but can in fact be exaggerated in amplitude, as indicated by two-headed arrows in red in Figs 2 and S4. Hence those physiological fluctuations that may be less relevant with the direct interaction between cerebral hemodynamic responses and $O_2$ or $CO_2$ would be larger for SI than for bER. We believe that is one of the explanations for why bER is superior to SI in its correlation with CBF (S4 Fig).

Another difference is that Bruce et al. [66] studied the correlation between CBF and the ratio $P_{ET}CO_2/P_{ET}O_2$ within a single prolonged period (~93 seconds) of breath holding, i.e. from the onset to the end of breath hold only. In our study, we evaluated the relationship between CBF and bER throughout the duration of six breath holding epochs (~30 seconds) and free breathing periods (~60–90 seconds). We took into account the long delayed cerebro-vascular response to breath hold as CBF continues to increase shortly after the breath holding period before it slowly returns to baseline during the free breathing period. Our team reported this delayed CBF response in our other TCD study on breath hold in 2009 [67] where we showed that the time duration of CBFv response to breath hold could last twice as long as the time duration of breath hold [67]. Another team also reported the same delayed CBF response to breath hold in an MRI study [10]. That is one reason why our use of bER is expected to be superior to the use of SI developed by Bruce et al. [66] in characterizing the CBF changes in response to breath hold. The stronger positive correlation of CBFv with bER than with SI suggests that there is synergistic behavior of $O_2$ and $CO_2$ in bER that goes beyond what SI does.

## Conclusion

Independent of our current knowledge to completely clarify why bER offers the strongest association with cerebral hemodynamic responses to breath holding, we succeeded in showing that bER was superior to $\Delta PCO_2$ or $P_{ET}CO_2$ to characterize cerebral hemodynamic responses under breath hold challenge for the CVR evaluation. RGE metrics of both $\Delta PO_2$ and $\Delta PCO_2$ should always be acquired for CVR evaluation instead of acquiring $P_{ET}CO_2$ data alone. In addition to offering alternative approach of CVR evaluation for patients who are not eligible for exogenous $CO_2$ challenge, the association between bER and cerebral hemodynamic responses also provides a novel insight in the study of brain-body interaction. Future studies would be required to clarify the underlying mechanisms for the relationship between dynamic bER and cerebral hemodynamic response to breath holding. Studies to quantify the relationship between changes in cerebral hemodynamic responses during breath holding and changes in bER in a large cohort of subjects and patients would be helpful to explore the effects on CVR by various disorders including respiratory or cerebral diseases with neurovascular deficits.

## Supporting information

**S1 Fig. Definition of end inspiration and end expiration on the time series of RGE metrics and the correlations among breath-by breath RGE matrices.** (A) A segment of 80-second time series of $\Delta CBFv$ in left MCA and physiological changes including breath-by-breath bER, $\Delta PO_2$, $\Delta PCO_2$, $P_{ET}O_2$ and $P_{ET}CO_2$ measured by gas analyzers and respiration time series (Resp) measured by respiratory bellow in a representative subject under breath hold challenge in TCD session. Open circles represent end expiration while closed circles represent end inspiration in resting phase or onset of expiration at the end of breath hold epoch. Positive phases with deflection above zero on the respiration time series represent inspiration and negative phases with deflection below zero represent expiration. The inspiratory and expiratory phases of each respiratory cycle on the time series of $P_{ET}O_2$ and $P_{ET}CO_2$ are verified by those on respiration time series. The timing for open (end expiration) and closed (end inspiration) circles in

green is the same as those in red and blue. (B) Correlations among breath-by breath respiratory matrices (bER, $\Delta PO_2$, $\Delta PCO_2$, ToB, $P_{ET}O_2$ and $P_{ET}CO_2$) in all subjects who participated in TCD sessions (n = 12), and (C) those who participated in MRI sessions (n = 16). Each gray circle represents the Pearson's correlation coefficient from the correlation analysis of the time series of parameter pair shown on x-axis for each subject. The thick middle horizontal line, the box and the vertical rod represent the mean, 95% confidence interval and standard deviation of the group data respectively. The time series of bER had stronger correlation with that of $\Delta PO_2$ than $\Delta PCO_2$, although both $\Delta PO_2$ and $\Delta PCO_2$ contributed to changes of bER. The correlation coefficients from $\Delta PO_2$ vs $\Delta PCO_2$ varied from 0.6 to 0.9 in TCD sessions and from 0.4 to 0.9 in MRI sessions, suggesting that $\Delta PO_2$ and $\Delta PCO_2$ are not necessarily redundant. The difference in the ranges of correlation strength found between TCD and MRI sessions may be due to the difference in posture of the subjects, where the subjects were in erect seated position in TCD sessions and they were in supine position in MRI sessions.
(TIF)

**S2 Fig. Wavelet transform coherence analysis between bER and $\Delta CBFv$ in a representative subject.** (A) Time series of bER and $\Delta CBFv$ measured in right MCA in a representative subject under breath hold challenge. (B) The squared wavelet coherence between these two time series. Squared wavelet coherence is plotted with x-axis as time and y-axis as scale which has been converted to its equivalent Fourier period. The magnitude of wavelet transform coherence ranges between 0 and 1, where warmer color represents stronger coherence and cooler color represents weaker coherence. Areas inside the 'cone of influence', which are locations in the time-frequency plane where edge effects give rise to lower confidence in the computed values, are shown in faded color outside of the conical contour. The statistical significance level of the wavelet coherence is estimated using Monte Carlo methods and the 5% significance level against red noise is shown as thick contour. The phase angle between the two time series at particular samples of the time-frequency plane is indicated by an arrow (rightward pointing arrows indicate that the time series are in phase or positively correlation, leftward pointing arrows indicate anticorrelation and the downward pointing arrows indicate phase angles of $\pi/2$). There are four different ranges of phase lags: $0+\pi/2$, $0-\pi/2$, $\pi-\pi/2$, and $\pi+\pi/2$. (C) Time-averaged coherences at four different phase lags of $0+\pi/2$, $0-\pi/2$, $\pi-\pi/2$, and $\pi+\pi/2$. At each phase lag range, time-averaged coherence was defined as the total significant coherence at each scale where the wavelet coherence magnitude exceeded 95% significance level, normalized by the maximum possible coherence outside the cone of influence, i.e. inside the conical contour, at that particular scale and phase lag range.
(TIF)

**S3 Fig. Coherence between time series of RGE metrics and cerebral hemodynamic response at four different phase lags ($0+\pi/2$, $0-\pi/2$, $\pi-\pi/2$, and $\pi+\pi/2$).** The mean time-averaged coherence between time series of respiratory metrics and cerebral hemodynamic responses ($\Delta CBFv$ in LMCA and RMCA, and $\Delta BOLD$ in LGM, RGM, LWM and RWM) at four different phase lags ($0+\pi/2$, $0-\pi/2$, $\pi-\pi/2$, and $\pi+\pi/2$) for the subjects included in the TCD sessions (n = 12) and in the MRI sessions (n = 16). Color shaded areas represent SEM. Comparing with $\Delta PO_2$, $\Delta PCO_2$ and ToB, the total time-averaged coherence between bER and cerebral hemodynamic responses was found to be significantly stronger between 0.008Hz (1/128 seconds) and 0.03Hz (1/32 seconds). The strong mean time-averaged coherence between respiratory metrics and cerebral hemodynamic responses were found at phase lag of $0+\pi/2$.
(TIF)

**S4 Fig. Time series of cerebral hemodynamic responses, bER and $P_{ET}CO_2/P_{ET}O_2$ in a representative subject under breath hold challenge.** Time series of $\Delta$CBFv in left MCA, bER and $P_{ET}CO_2/P_{ET}O_2$ in the same representative subject in Fig 2A under breath hold challenge in TCD session. Shaded areas represent breath hold periods. The time series of bER followed closely to the $\Delta$CBFv changes, while $P_{ET}CO_2/P_{ET}O_2$ did not follow $\Delta$CBFv changes some time during the challenge as indicated by two-headed arrow in red. In the time period between 400 and 580 seconds (as indicated by two-headed arrow) when the subject had shallow breathing, the amplitude of $P_{ET}CO_2/P_{ET}O_2$ decreased significantly in comparison with that of bER. This may be attributed to the different property of gas measurements where the time series of $\Delta PO_2$ and $\Delta PCO_2$ oscillated in phase while those of $P_{ET}O_2$ and $P_{ET}CO_2$ oscillated out of phase (Fig 2A). $P_{ET}CO_2$ was decreased and $P_{ET}O_2$ was increased by shallow breathing, resulting in a significant decrease in $P_{ET}CO_2/P_{ET}O_2$.
(TIF)

**S1 Table. Correlation among RGE metrics.** Strength of correlation indicated by Pearson's correlation coefficients among respiratory metrics including bER, $\Delta PO_2$, $\Delta PCO_2$, ToB, $P_{ET}O_2$ and $P_{ET}CO_2$ in all subjects who participated in TCD sessions (n = 12), and those who participated in MRI sessions (n = 16). The time series of bER had stronger correlation with that of $\Delta PO_2$ than $\Delta PCO_2$, although both $\Delta PO_2$ and $\Delta PCO_2$ contributed to changes of bER. The correlation coefficients from $\Delta PO_2$ vs $\Delta PCO_2$ varied from 0.6 to 0.9 in TCD sessions and from 0.4 to 0.9 in MRI sessions, suggesting that $\Delta PO_2$ than $\Delta PCO_2$ are not necessarily redundant.
(DOCX)

**S2 Table.** A. Correlation between RGE metrics and $\Delta$CBFv in TCD sessions. Strength of correlation indicated by Pearson's correlation coefficients between $\Delta$CBFv and RGE metrics including bER, $\Delta PO2$, $\Delta PCO2$ and ToB (n = 12). Numbers in brackets next to Pearson's correlation coefficients indicate p values from individual correlation analyses. The bottom row shows the mean values of Fisher Z scores transformed from Pearson's correlation coefficients in groups. Numbers in brackets next to mean Fisher Z scores indicate p values in the paired comparisons. The correlation between $\Delta$CBFv and bER was significantly larger than those of the correlation between $\Delta$CBFv and the other respiratory metrics in the paired comparisons (p<0.001). bER is the only parameter that consistently showed significantly high correlation with the $\Delta$CBFv measured in LMCA and RMCA. B. Correlation between RGE metrics and $\Delta$BOLD in MRI sessions. Strength of correlation indicated by Pearson's correlation coefficients between $\Delta$BOLD and RGE metrics including bER, $\Delta PO2$, $\Delta PCO2$ and ToB (n = 16). Numbers in brackets next to Pearson's correlation coefficients indicate p values from individual correlation analyses. The bottom row shows the mean values of Fisher Z scores transformed from Pearson's correlation coefficients in groups. Numbers in brackets next to mean Fisher Z scores indicate p values in paired comparisons. The correlation between $\Delta$BOLD and bER was significantly larger than those of the correlation between $\Delta$BOLD and the other respiratory metrics in the paired comparisons (p<0.001). bER is the only parameter that consistently showed significantly high correlation with the $\Delta$BOLD measured in LGM, RGM, LWM and RWM.
(DOCX)

## Author Contributions

**Conceptualization:** Suk-tak Chan, Kenneth K. Kwong.

**Data curation:** Suk-tak Chan, Karleyton C. Evans, Kenneth K. Kwong.

**Formal analysis:** Suk-tak Chan.

**Funding acquisition:** Bruce R. Rosen.

**Investigation:** Suk-tak Chan, Karleyton C. Evans, Tian-yue Song, Juliette Selb, Kenneth K. Kwong.

**Methodology:** Suk-tak Chan, Karleyton C. Evans, Kenneth K. Kwong.

**Project administration:** Kenneth K. Kwong.

**Resources:** Bruce R. Rosen, Yong-ping Zheng, Andrew Ahn, Kenneth K. Kwong.

**Software:** Andre van der Kouwe.

**Supervision:** Bruce R. Rosen, Kenneth K. Kwong.

**Writing – Original Draft:** Suk-tak Chan.

**Writing – Review & Editing:** Suk-tak Chan.

**Writing – review & editing:** Tian-yue Song, Bruce R. Rosen, Yong-ping Zheng, Kenneth K. Kwong.

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
