## [Decision Letter · Decision Letter 0]

19 Dec 2019

PONE-D-19-31336

Cerebrovascular reactivity assessment with O2-CO2 exchange ratio under brief breath hold challenge

PLOS ONE

Dear Chan,

Thank you for submitting your manuscript to PLOS ONE. After careful consideration, we feel that it has merit but does not fully meet PLOS ONE’s publication criteria as it currently stands. Therefore, we invite you to submit a revised version of the manuscript that addresses the points raised during the review process.

We would appreciate receiving your revised manuscript by Feb 02 2020 11:59PM. To enhance the reproducibility of your results, we recommend that if applicable you deposit your laboratory protocols in protocols.io, where a protocol can be assigned its own identifier (DOI) such that it can be cited independently in the future. For instructions see: http://journals.plos.org/plosone/s/submission-guidelines#loc-laboratory-protocols

We look forward to receiving your revised manuscript.

Kind regards,

Shigehiko Ogoh

Academic Editor

PLOS ONE

Additional Editor Comments:

Chan et all investigated novel ways of quantifying cerebrovascular reactivity in response to acute perturbations in blood gases. This study was basically a methods study, investigating new metrics and comparing measurement techniques. However, the background, rational and hypothesis difficult to follow, and would suggest the authors more clearly lay out their study for the sake of the reader.

Major Comments

In both the abstract and the introduction, it it difficult to track the rationale and study design. It felt like a bit of a post-hoc packaging decision following data collection of two studies, as opposed to an a priori developed study to investigate a specific question(s). For example, a well-written introduction would lay out related background information, with one main topic per paragraph, leading intuitively to a clear rationale(s) (i.e., what is not known), then aims and specific directional hypotheses. Instead, the authors pepper up to five separate hypotheses throughout the introduction. The introduction only ends with a statement about potential significance. I feel the introduction needs a major re-write for clarity in relevant background, rationale, aim and hypothesis. I recognize that methods papers are difficult to frame in the same way as experimental papers, but it would be helpful if the authors more clearly outlined the rationale, with references, why they sought to develop novel metrics.

The authors are making a claim that they have developed a novel metric to assess CVR. However, I would question both the relevance claim and the novelty claim. The bER metrics is basically just a breath-by-breath inverse of RER, as they indicate, which is interesting. I can understand the need for higher temporal resolution on a breath-by-breath basis, but the inverse nature of their metric is confusing. RER is quantified as a given amount of CO2 created (and expired) for a given amount of O2 consumed, and tells us something about what fuel is being utilized to the metabolic rate. Indeed, on the point that both CO2 and O2 are relevant to cerebrovascular response to breath holding, I agree, as do others. I find it inexplicable to inverse this metric. Why express bER as O2/CO2? Further, in terms of both respiratory and cerebrovascular responses to blood gases, CBF is directly proportional to changes in CO2, and inversely proportional to O2 and/or SaO2. Thus, this reviewer feels that the bER being arranged as inverse of both RER and known physiological responses unjustifiable.

I also question the novelty of this app[roach. Indeed, Luacs et al 2011 (J Physiol) assessed CBF with a similar O2/CO2 relationship during ascent to high altitude. I feel they got the relationship backward, as per my comments above. In addition, Bruce et al 2016 (Exp Physiol) developed a technique to assess CBF responses to breath holding taking into account instantaneous changes in CO2 and O2, indexing CBV against a breath-by-breath stimulus index (SI; CO2/O2). They interpolated these gases from those at the beginning and end of a breath hold, which is basically your delta measures, I believe. Lafave et al 2019 (EJAP) utilized this SI during ascent to high altitude.

As you measured MAP, if you wanted to go all the way, why no express breath-by-breath cerebrovascular conductance (CVC = CBV/MAP).

Minor Comments

Please refer to TCD metrics as cerebral blood velocity or CBV, throughout. There is no such thing as cerebral blood flow velocity.

4. Thank you for stating the following in the Financial Disclosure section:

'This research was carried out in whole at the Athinoula A. Martinos Center for Biomedical Imaging at the Massachusetts General Hospital, using resources provided by the Center for Functional Neuroimaging Technologies, P41EB015896, a P41 Biotechnology Resource Grant supported by the National Institute of Biomedical Imaging and Bioengineering (NIBIB), National Institutes of Health, as well as the Shared Instrumentation Grant S10RR023043.  This work was also supported, in part, by NIH-K23MH086619. '

We note that one or more of the authors are employed by a commercial company:  Biogen Inc.

Reviewers' comments:

Reviewer's Responses to Questions

**Comments to the Author**

1. Is the manuscript technically sound, and do the data support the conclusions?

Reviewer #1: Yes

2. Has the statistical analysis been performed appropriately and rigorously? 

Reviewer #1: Yes

3. Have the authors made all data underlying the findings in their manuscript fully available?

Reviewer #1: Yes

4. Is the manuscript presented in an intelligible fashion and written in standard English?

Reviewer #1: Yes

5. Review Comments to the Author

Reviewer #1: Breath hold challenge is a simple vasoactive stimulus for the assessment of cerebrovascular reactivity (CVR) that is used in the clinic as well as exogenous CO2 challenge (hypercapnia test). The authors of the manuscript have demonstrated that the cerebrovascular response to brief breath hold hypercapnia test, used in neuro-intensive care, are coupled not only with the increased partial pressure of carbon dioxide (as it was believed before) but also with a decrease in the partial pressure of oxygen. These findings suggest that mild hypercapnia could increase the sensitivity of the CBF response to a very mild level of hypoxia and the ranges of mild PO2 and PCO2 changes reported are achievable by breath hold. This is of great importance as it means that the physiological mechanisms of cerebrovascular changes underlying breath hold and exogenous CO2 challenges are potentially different. The authors found that the breath-by-breath O2-CO2 exchange ratio (bER), namely the ratio of changes in PO2 (∆PO2) to changes in PCO2 (∆PCO2) between end inspiration and end expiration, was superior to either ∆PO2 or ∆PCO2 alone in coupling with the changes of CBFv and BOLD signals under breath hold challenge. Thus, bER would be able to better characterize CVR under breath hold challenge in the regression model without creating a problem of collinearity.

6. PLOS authors have the option to publish the peer review history of their article (what does this mean?). If published, this will include your full peer review and any attached files.

Reviewer #1: No

---

## [Author Response · Author response to Decision Letter 0]

23 Jan 2020

Our team would like to thank for the useful comments and suggestions from the Reviewers and Editor. Please find our responses (in blue) to the Reviewers’ comments below.

Major Comments:

1. In both the abstract and the introduction, it is difficult to track the rationale and study design. It felt like a bit of a post-hoc packaging decision following data collection of two studies, as opposed to an a priori developed study to investigate a specific question(s). For example, a well-written introduction would lay out related background information, with one main topic per paragraph, leading intuitively to a clear rationale(s) (i.e., what is not known), then aims and specific directional hypotheses. Instead, the authors pepper up to five separate hypotheses throughout the introduction. The introduction only ends with a statement about potential significance. I feel the introduction needs a major re-write for clarity in relevant background, rationale, aim and hypothesis. I recognize that methods papers are difficult to frame in the same way as experimental papers, but it would be helpful if the authors more clearly outlined the rationale, with references, why they sought to develop novel metrics.

Responses: The abstract and introduction have been rewritten based on the reviewer’s comment and suggestion. We removed the numerous sub-hypotheses distributed throughout the Introduction and raised a single hypothesis in the last paragraph of the Introduction stating that “mild hypoxia and hypercapnia work synergistically to increase CBF under breath hold challenge”. We listed our major objective after the hypothesis. Each of the preceding paragraphs presented specific background information as well as its associated rationale (i.e. what is not known).

2. The authors are making a claim that they have developed a novel metric to assess CVR. However, I would question both the relevance claim and the novelty claim. The bER metrics is basically just a breath-by-breath inverse of RER, as they indicate, which is interesting. I can understand the need for higher temporal resolution on a breath-by-breath basis, but the inverse nature of their metric is confusing. RER is quantified as a given amount of CO2 created (and expired) for a given amount of O2 consumed and tells us something about what fuel is being utilized to the metabolic rate. Indeed, on the point that both CO2 and O2 are relevant to cerebrovascular response to breath holding, I agree, as do others. I find it inexplicable to inverse this metric. Why express bER as O2/CO2? 

Responses: We will first answer the question of bER vs. breath-by-breath RER. Both bER and RER describe the process of respiratory gas exchange. We reported that bER, but not RER, was compatible with a positive correlation with CBF (Fig 3 in original manuscript). Our data show that CBF is positively correlated with bER but negatively correlated with RER. We selected bER because a respiratory gas stimulus is conventionally expected to be positively correlated with a CBF response. 

CBF is described by hemodynamic surrogates CBFv and BOLD in this manuscript. Our explanation to use cerebral blood flow velocity (CBFv) instead of cerebral blood velocity (CBV) is given below in our first response to minor comments. 

For more clarity, it helps to repeat bER = ΔPO2/ΔPO2. ΔPO2 = (inspired PO2 – expired PO2) and ΔPCO2 = (expired PCO2 – inspired PCO2). ΔPO2 and ΔPCO2 are physiologically related to the change of gas partial pressure in systemic O2 uptake and that in CO2 release respectively.

3. Further, in terms of both respiratory and cerebrovascular responses to blood gases, CBF is directly proportional to changes in CO2, and inversely proportional to O2 and/or SaO2. Thus, this reviewer feels that the bER being arranged as inverse of both RER and known physiological responses unjustifiable.

I also question the novelty of this approach. Indeed, Luacs et al 2011 (J Physiol) assessed CBF with a similar O2/CO2 relationship during ascent to high altitude. I feel they got the relationship backward, as per my comments above. In addition, Bruce et al 2016 (Exp Physiol) developed a technique to assess CBF responses to breath holding taking into account instantaneous changes in CO2 and O2, indexing CBV against a breath-by-breath stimulus index (SI; CO2/O2). They interpolated these gases from those at the beginning and end of a breath hold, which is basically your delta measures, I believe. Lafave et al 2019 (EJAP) utilized this SI during ascent to high altitude.

Responses: We will give a fairly long response below to the reviewer’s comment that “cerebral blood flow (CBF) is directly proportional to changes in CO2, and inversely proportional to O2 and/or SaO2,” and his associated citation of the ‘stimulus index’ (SI: PETCO2/PETO2) developed by Bruce et al. [1]. Both Bruce et al. (using SI) and our team (using bER) agree upon the influence of O2 and CO2 on CBFv change in breath hold. It is natural to wonder whether SI and bER behave in a similar way. But there are surprisingly quite a few differences between SI and bER. In our original manuscript, we thought that going into an extensive discussion about the numerous differences between Bruce et al. and us belonged to another study as we wanted to focus on characterizing and clarifying the interaction between respiratory gas exchange and CBF in breath hold. Since the reviewer is highly concerned about our work and that of Bruce et al, we now include discussion about the differences between SI and bER at the end of our revised Discussion section as well as in the Supplementary Information. 

The many differences between bER and SI in characterizing CBF changes in breath hold will be discussed point-by-point as follows.

a) Is bER closely similar to SI in its interaction with CBF in terms of the breath hold protocol? The short answer is no. Does CBF show a stronger correlation with bER than with SI? The answer is yes. We calculated the time courses of SI (i.e. PETCO2/PETO2) using the data values on the time courses of PETCO2 and PETO2 from our representative subject in Fig 2A of the original manuscript, and correlated the values of PETCO2/PETO2 with changes of CBFv. The results are shown in S4 Fig (Supplementary Information). Our results indicate that bER is different from SI in its interaction with CBF. First, the correlation coefficient between bER and SI is 0.7 instead of being close to 1. Secondly, CBFv of the same subject correlates better with bER than SI. The stronger positive correlation of CBFv with bER than with SI suggests that there is synergistic behavior of O2 and CO2 in bER that goes beyond what SI does.

b) A feature that contributes to the differences between bER and SI could be the direction/phase of oscillations of gas measurements. As shown in Fig 2 of the original manuscript, the time courses of PETO2 and PETCO2 oscillate out of phase, while ΔPO2 oscillates in phase with ΔPCO2. It is important to recognize that many less interesting physiological signals (e.g. respiratory fluctuation) are shared and carried by both O2 and CO2 time courses. With ΔPO2 being in phase with ΔPCO2, those shared physiological signals would be largely factored out in the ratio ΔPO2/ΔPCO2 (bER). With PETO2 and PETCO2 being out of phase, those less interesting physiological signals do not get factored out by SI but can in fact be exaggerated in amplitude, as indicated by two-headed arrows in red in Fig 2 and in S4 Fig (Supplementary Information). Hence those physiological fluctuations that may be less relevant with the direct interaction between cerebral hemodynamic influence and O2 or CO2 would be larger for SI than for bER. We believe that is one of the explanations for why bER is superior to SI in its correlation with CBF. 

For more clarity, we repeat here some of the relationship between ΔPO2, PETO2, ΔPCO2 and PETCO2. If inspired PO2 is assumed to be constant, ΔPO2 increases with decrease in PETO2. Since the PCO2 in inspired air is almost zero, ΔPCO2 is basically equivalent to PETCO2. The oscillations of ΔPO2, ΔPCO2 and PETCO2 are all in phase while PETO2 is out of phase.

c) Bruce et al. studied the correlation between CBF and SI only within the period of breath holding (i.e. from the onset to the end of breath hold) by interpolating the PETCO2 and PETO2 measurements. In our study, we evaluated the relationship between CBF and bER throughout the duration of both breath holding and free breathing periods. The reason is that we take into account the long delayed cerebrovascular response to breath holding as CBF continues to increase shortly after the breath holding period before it slowly returns to baseline during the free breathing period. Our team reported this delayed CBF response in our other TCD study on breath hold in 2009 [2] where we showed that the time duration of CBFv response to breath hold could last twice as long as the time duration of breath hold (Figure 2 in Chan et al. [2]). Other teams also reported the same delayed CBF response to breath hold in an MRI study [3]. That is one reason why our use of bER is expected to be superior to the use of SI developed by Bruce et al. in characterizing the CBF changes in response to breath hold. 

d) Different from previous breath hold TCD studies which include the work by Bruce et al, we provided also MRI results to study BOLD responses in different brain regions as well. We are therefore able to study relationship between bER and cerebrovascular responses, both in the blood supply to the major cerebral territories using TCD and in regional BOLD responses using MRI. 

e) Bruce et al. included only one prolonged epoch of breath holding and the duration of breath holding lasted much more than one minute (mean psychological break-point = 93.4 sec, Table 1 in Bruce et al. [1]). In our study, we had six epochs of 30-second breath holding (Fig 1 in original manuscript) based on the consideration of the tolerability of normal subjects as well as patients. Our approach is more realistic for clinical purpose. 

f) Bruce et al. accepted in its Introduction section that both O2 and CO2 would have an effect on CBF during breath hold but did not raise any discussion about the long held claim in the literature that arterial PO2 needs to be around 50mmHg [4, 5] before significant CBF increase could be expected. Our manuscript explored the model on how O2 works synergistically with CO2 in raising CBF in breathhold even at relatively low level of ΔPO2. We specifically pointed out not only CBFv correlates best with bER but CBFv also correlates better with ΔPO2 than with ΔPCO2 (Fig 3 in original manuscript). The role of O2 was less explored in Bruce et al. 

g) Our choice of using ΔPO2 instead of end-tidal PO2 (PETO2) helps us to highlight the role of O2-CO2 exchange. The reason is that ΔPO2 and ΔPCO2 are physiologically related to the change of gas partial pressure in systemic O2 uptake and that in CO2 release respectively. In fact ΔPO2, and not PETO2, is the proper term used by Fenn et al. [6] and Ferretti [7] to express respiratory gas exchange in the alveolar air equation. 

Since our objective was to measure how CBFv is related to the process of respiratory gas exchange, we used the proper term ΔPO2 and not PETO2. 

Incidentally, RER in the alveolar air equation is usually displayed in the form of 1/RER. It is equivalent to replace 1/RER by bER averaged over time. 

4. As you measured MAP, if you wanted to go all the way, why no express breath-by-breath cerebrovascular conductance (CVC = CBV/MAP).

Responses: The effect of mild MAP change on CBF change relates to autoregulation. At a short breath hold duration of 30 seconds with relatively mild MAP change, we prefer to focus more on the effect of of respiratory gas exchange of O2 and CO2 on CBF change in this manuscript. 

Minor Comments:

1. Please refer to TCD metrics as cerebral blood velocity or CBV, throughout. There is no such thing as cerebral blood flow velocity.

Responses: In biology community, we understand that the velocity of blood flow is usually referred to ‘blood velocity’. However, to ensure the precise communication in the imaging and neuroscience communities, we used the standard terminology of ‘blood flow velocity’ as in one of the first papers by Aaslid et al. [8] on using TCD method to measure the flow velocity in cerebral arteries. In addition, CBV in neuroimaging is commonly referred to ‘cerebral blood volume’. 

General Responses: Our responses to the Reviewer’s comments on SI had been incorporated into the revised Discussion section as one of the paragraphs under the subheading of “Why was the ratio of delta PO2 to delta PCO2 used, and not the product?”. 

 

http://www.journals.plos.org/plosone/s/file?id=wjVg/PLOSOne_formatting_sample_main_body.pdf and

http://www.journals.plos.org/plosone/s/file?id=ba62/PLOSOne_formatting_sample_title_authors_affiliations.pdf

Responses: We revise the areas which do not match PLOS ONE’s style requirements. The affiliation of the second author (Karl Evans) is changed back to the affiliation when the study was done, i.e. Department of Psychiatry, Massachusetts General Hospital. His current address (Biogen, Inc) is included.

2. We note that you have indicated that data from this study are available upon request. PLOS only allows data to be available upon request if there are legal or ethical restrictions on sharing data publicly. For information on unacceptable data access restrictions, please see http://journals.plos.org/plosone/s/data-availability#locunacceptable-data-access-restrictions.

a) If there are ethical or legal restrictions on sharing a de-identified data set, please explain them in detail (e.g.,data contain potentially identifying or sensitive patient information) and who has imposed them (e.g., an ethics committee). Please also provide contact information for a data access committee, ethics committee, or other institutional body to which data requests may be sent.

Responses: As the data for individual subjects were included in the Figures, Tables and Supporting Information, we amend the data availability statement as follow:

‘All relevant data are within the paper and its Supporting Information files.’

Responses: My existing ORCID has been added in ‘Update My Information’ in PLOS Editorial Manager.

4. Thank you for stating the following in the Financial Disclosure section:

'This research was carried out in whole at the Athinoula A. Martinos Center for Biomedical Imaging at the Massachusetts General Hospital, using resources provided by the Center for Functional Neuroimaging Technologies, P41EB015896, a P41 Biotechnology Resource Grant supported by the National Institute of Biomedical Imaging and Bioengineering (NIBIB), National Institutes of Health, as well as the Shared Instrumentation Grant S10RR023043. This work was also supported, in part, by NIH-K23MH086619. '

We note that one or more of the authors are employed by a commercial company: Biogen Inc.

Responses: The first two funders in the statement mainly supported the MRI system for data collection, computer servers for data analysis and storage. They did not provide salary support to the authors specifically for this study. The last funder supported part of the respiratory instrumentation and disposables. Although these funders did not contribute in the study design, publish and preparation of manuscript, they contributed the equipment for data collection and analysis. The Funding Statement is amended as follow:

'This research was carried out in whole at the Athinoula A. Martinos Center for Biomedical Imaging at the Massachusetts General Hospital, using resources provided by the Center for Functional Neuroimaging Technologies, P41EB015896, a P41 Biotechnology Resource Grant supported by the National Institute of Biomedical Imaging and Bioengineering (NIBIB), National Institutes of Health, as well as the Shared Instrumentation Grant S10RR023043. This work was also supported, in part, by NIH-K23MH086619. The funders had no role in study design, decision to publish, or preparation of the manuscript'

The affiliation of the second author (Karl Evans) is changed back to the affiliation when the study was done, i.e. Department of Psychiatry, Massachusetts General Hospital. His current address (Biogen, Inc) is included. Biogen, Inc. is only Karl’s current affiliation and does not have any role in the study. 

Responses: The Competing Interests Statement is included in the cover letter as follow:

‘The authors have declared that no competing interests exist.’

Responses: The oxygen saturation data are not core part of this study. The statement is removed.

 

Reviewer #1: Breath hold challenge is a simple vasoactive stimulus for the assessment of cerebrovascular reactivity (CVR) that is used in the clinic as well as exogenous CO2 challenge (hypercapnia test). The authors of the manuscript have demonstrated that the cerebrovascular response to brief breath hold hypercapnia test, used in neuro-intensive care, are coupled not only with the increased partial pressure of carbon dioxide (as it was believed before) but also with a decrease in the partial pressure of oxygen. These findings suggest that mild hypercapnia could increase the sensitivity of the CBF response to a very mild level of hypoxia and the ranges of mild PO2 and PCO2 changes reported are achievable by breath hold. This is of great importance as it means that the physiological mechanisms of cerebrovascular changes underlying breath hold and exogenous CO2 challenges are potentially different. The authors found that the breath-by-breath O2-CO2 exchange ratio (bER), namely the ratio of changes in PO2 (ΔPO2) to changes in PCO2 (ΔPCO2) between end inspiration and end expiration, was superior to either ΔPO2 or ΔPCO2 alone in coupling with the changes of CBFv and BOLD signals under breath hold challenge. Thus, bER would be able to better characterize CVR under breath hold challenge in the regression model without creating a problem of collinearity.

Responses: Our team appreciates the comments from Reviewer #1.

 

References

1. Bruce CD, Steinback CD, Chauhan UV, Pfoh JR, Abrosimova M, Vanden Berg ER, et al. Quantifying cerebrovascular reactivity in anterior and posterior cerebral circulations during voluntary breath holding. Exp Physiol. 2016;101(12):1517-27.

2. Chan ST, Tam Y, Lai CY, Wu HY, Lam YK, Wong PN, et al. Transcranial Doppler study of cerebrovascular reactivity: are migraineurs more sensitive to breath-hold challenge? Brain Res. 2009;1291:53-9.

3. Murphy K, Harris AD, Wise RG. Robustly measuring vascular reactivity differences with breath-hold: normalising stimulus-evoked and resting state BOLD fMRI data. NeuroImage. 2011;54(1):369-79.

4. Lassen NA. Cerebral blood flow and oxygen consumption in man. Physiological reviews. 1959;39(2):183-238.

5. Hoiland RL, Bain AR, Rieger MG, Bailey DM, Ainslie PN. Hypoxemia, oxygen content, and the regulation of cerebral blood flow. American journal of physiology Regulatory, integrative and comparative physiology. 2016;310(5):R398-413.

6. Fenn WO, Rahn H, Otis AB. A theoretical study of the composition of the alveolar air at altitude. Am J Physiol. 1946;146:637-53.

7. Ferretti G. Energetics of Muscular Exercise. Switzerland: Springer International Publishing; 2015.

8. Aaslid R, Markwalder TM, Nornes H. Noninvasive transcranial Doppler ultrasound recording of flow velocity in basal cerebral arteries. J Neurosurg. 1982;57(6):769-74.

---

## [Decision Letter · Decision Letter 1]

28 Feb 2020

Cerebrovascular reactivity assessment with O2-CO2 exchange ratio under brief breath hold challenge

PONE-D-19-31336R1

Dear Dr. Chan,

We are pleased to inform you that your manuscript has been judged scientifically suitable for publication and will be formally accepted for publication once it complies with all outstanding technical requirements.

With kind regards,

Shigehiko Ogoh

Academic Editor

PLOS ONE

Additional Editor Comments (optional):

Thank you for the revised manuscript.　Both reviewers are satisfied with your response.

Reviewers' comments:

Reviewer's Responses to Questions

**Comments to the Author**

1. If the authors have adequately addressed your comments raised in a previous round of review and you feel that this manuscript is now acceptable for publication, you may indicate that here to bypass the “Comments to the Author” section, enter your conflict of interest statement in the “Confidential to Editor” section, and submit your "Accept" recommendation.

Reviewer #2: All comments have been addressed

2. Is the manuscript technically sound, and do the data support the conclusions?

Reviewer #2: Yes

3. Has the statistical analysis been performed appropriately and rigorously? 

Reviewer #2: Yes

4. Have the authors made all data underlying the findings in their manuscript fully available?

Reviewer #2: Yes

5. Is the manuscript presented in an intelligible fashion and written in standard English?

Reviewer #2: Yes

6. Review Comments to the Author

Reviewer #2: Thank you to the authors for their thoughtful responses to my queries.

The issue of how to quantify CBF responses to a breath hold stimulus is difficult, and the authors have carried out an interesting study (using two metrics of CBF), to characterize a novel metric that takes into account changes in both O2 and CO2.

They have addressed my queries about organization of the introduction and their quantification technique adequately, and they explain clearly in the manuscript why the RER, SI and bER are different with respect to CBF responses, as they did in the rebuttal to me. This reconciliation between analysis techniques is useful and important.

Normally, the hypothesis is the last sentence in a manuscript. The potential significance can be left to the discussion.

Congratulations on an interesting study.

Make sure you cite the justification of why BP was unchanged during your breath hold given other work:

https://www.ncbi.nlm.nih.gov/pubmed/21521758

A new study that you may be interested in was just published:

https://www.ncbi.nlm.nih.gov/pubmed/32083357

Other useful Refs:

https://www.ncbi.nlm.nih.gov/pubmed/24081155

https://www.ncbi.nlm.nih.gov/pubmed/22961068

7. PLOS authors have the option to publish the peer review history of their article (what does this mean?). If published, this will include your full peer review and any attached files.

Reviewer #2: No

---

## [Editor Report · Acceptance letter]

5 Mar 2020

PONE-D-19-31336R1 

Cerebrovascular reactivity assessment with O2-CO2 exchange ratio under brief breath hold challenge 

Dear Dr. Chan:

I am pleased to inform you that your manuscript has been deemed suitable for publication in PLOS ONE. Congratulations! Your manuscript is now with our production department. 

With kind regards,

on behalf of

Dr. Shigehiko Ogoh 

Academic Editor

PLOS ONE